# CARROT: A Cost Aware Rate Optimal Router

## Abstract

With the rapid growth in the number of Large Language Models (LLMs), there has been a recent interest in *LLM routing*, or directing queries to the cheapest LLM that can deliver a suitable response. We conduct a minimax analysis of the routing problem, providing a lower bound and finding that a simple router that predicts both cost and accuracy for each question can be minimax optimal. Inspired by this, we introduce CARROT, a Cost AwaRe Rate Optimal rouTer that selects a model based on estimates of the models' cost and performance. Alongside CARROT, we also introduce the Smart Price-aware ROUTing (SPROUT) dataset to facilitate routing on a wide spectrum of queries with the latest state-of-the-art LLMs. Using SPROUT and prior benchmarks such as Routerbench and open-LLM-leaderboard-v2 we empirically validate CARROT's performance against several alternative routers.

## 1 Introduction

Large language models (LLMs) have demonstrated the capability to effectively address a diverse array of tasks across academic, industrial, and everyday settings (Minaee et al., 2024). This continued success has catalyzed the rapid development of new LLMs tailored for both general and specialized applications (Myrzakhan et al., 2024). While this offers practitioners increased flexibility, the vast number of available options may pose a daunting challenge in their real-world deployment. Particularly, determining the optimal LLM for a given query remains a significant challenge. In a perfect world, all queries can be routed to the most powerful model, but for many, this may quickly become prohibitively expensive.

A common approach to address this issue is *routing* (Shnitzer et al., 2023; Hu et al., 2024; Ong et al., 2024; Jain et al., 2023; Šakota et al., 2024; Chen et al., 2022; Nguyen et al., 2025). There are two paradigms of routing; *non-predictive* routers repeatedly call LLMs and evaluate the responses to select the best one for a given query. Examples include Fusion of Experts (FoE) (Wang et al., 2023), FrugalGPT (Chen et al., 2024), and techniques that cascade answers from weak to strong LLMs (Yue et al., 2024). The obvious disadvantage of non-predictive routing is the required inference of many LLMs for all queries, even those that are not suitable for the task at hand. As a workaround, researchers have also considered *predictive routers*, which take LLM queries as inputs and output guesses at the most appropriate LLM. A key limitation of the prior literature on predictive routing is *the avoidance of the cost prediction problem* for text generation in unknown queries. For example, Shnitzer et al. (2023) only considers performance prediction. In another direction, RouteLLM (Ong et al., 2024) and RoRF (Jain et al., 2023) take a step forward and implicitly incorporate model cost by creating binary routers that select between a large, costly model and a cheap, small model. However, they do not predict the cost of individual queries and, as we shall see, the reduced flexibility of binary routing leads to performance degradation in practice. A recent work, Hu et al. (2024), introduces a router that considers cost and accuracy, but they assume that cost is constant across all questions in the dataset. The works Chen et al. (2022); Nguyen et al. (2025) consider dynamic vs. static cost prediction in routing, but their attention is limited to classification tasks; as the inference cost of LLMs is heavily dependent on the number of output tokens, studying this question for more open-ended prompts remains an important problem. Finally, Šakota et al. (2024) considers cost prediction, but their method does not generalize to unseen queries, undermining its use in more realistic applications. In summary,

they require prior knowledge of the test set queries, as test-time routing decisions are made by solving a Linear Program for each query. In contrast, we will introduce a router that can handle an unknown stream of questions at test time.

To quantify the importance of cost prediction in routing, we provide a minimax analysis of the routing problem. In Theorem 3.6, we establish a lower bound on minimax excess risk for any possible LLM router, in terms of the training sample size and certain quantities of the underlying prompt and model cost/accuracy distribution. Next, in Theorem 3.9 we show that a simple router based on predicting *both LLM cost and accuracy* from a given prompt can achieve the minimax lower bound established in Theorem 3.6. Collecting adequate data to train a router is challenging; for each LLM and every query one must collect a response *and* an evaluation from a judge of that response. By necessity, this collection process must include inference from closed source models. This makes the routing data gathering process expensive; the following informal Theorem emphasizes the importance of cost prediction in producing a router that makes the best use of this expensive data.

**Theorem 1.1** (Theorems 3.6 and 3.9 informal). *An LLM router that predicts both cost and accuracy for every question and all models in a family can achieve optimal statistical efficiency.*

Inspired by these findings (and to test them empirically), we introduce CARROT: a Cost AwaRe Rate Optimal rouTer and the Smart Price-aware ROUTing (SPROUT) dataset. CARROT utilizes a simple two-stage approach. We first attain an estimator for each of the metrics (*e.g.* cost and accuracy) for each model given a query, then we plug in these estimators into the formed risk function and select a model that minimizes the appropriate convex combination of the estimated metrics.

The key learning step is attaining these aforementioned predictors, and this is where SPROUT comes into play. SPROUT covers 14 state-of-the-art language models (*e.g.*, Llama-3-herd (Grattafiori et al., 2024), GPT-4o (Achiam et al., 2024), *etc.*) and approximately 45k prompts from 6 benchmarks covering RAG, science, reasoning, and GPT-4 generated user queries. For all models, we use zero-shot prompting and corresponding chat templates to represent practical use cases and collect input and output token counts to allow flexibility when studying cost-performance trade-offs. As a sneak peek, in Figure 1, we present the ratio of CARROT's performance to GPT-4o's (Achiam et al., 2024) on several key benchmarks across diverse use cases represented in SPROUT. At 30% of the cost, CARROT matches or exceeds the performance of GPT-4o on each benchmark.

### 1.1 Paper Outline

In Section 2, we introduce the routing problem and the "plug-in" approach to routing that CARROT utilizes. In Section 3, we provide our minimax analysis of the routing problem. In Section 4, we introduce SPROUT and empirically test the theoretical ideas discussed in the prior sections. To test CARROT's efficiency compared with prior routers, we utilize it to estimate the Pareto frontier of performance and cost trade-off on RouterBench(Hu et al., 2024), open-LLM-leaderboard-v2 (Fourrier et al., 2024), and our new SPROUTdataset.

### 1.2 Related literature

**Performance vs cost trade-off in LLM predictions.** Several recent studies have explored optimizing the cost and performance trade-offs in the implementation of large-language models (LLMs). LLM-BLENDER (Jiang et al., 2023) ensembles outcomes from multiple LLMs to select the best response. Frugal-ML, Frugal-GPT (Chen et al., 2020; 2024) and FrugalFoE Wang et al. (2023) employ an LLM cascade to sequentially query LLMs until a reliable response is found. AutoMix (Madaan et al., 2023) relies on a smaller model to self-verify its response before potentially considering a larger model. While these approaches rely on multiple LLM queries, our approach routes each query to a single LLM, an approach also considered in Hu et al. (2024). We complement these works by providing a statistically principled approach to learning this performance vs. cost trade-off.

**Ensemble learning.** The routing problem is closely related to ensemble learning that combines multiple models to obtain better performance. Classical ensemble methods include bagging (bootstrap aggregating), boosting, and stacking (model blending) (Breiman, 1996a;b; Freund et al., 1996; Friedman, 2001; Wolpert, 1992). Most of these works implicitly assume that the models in the ensemble have *similar expertise*, and thus it is beneficial to aggregate their predictions, whereas in our case, models may have *complementary expertise*, and averaging their outputs might be detrimental because most of them may not be suitable for an input. Therefore, we choose to predict using the model with the best outcome, rather than aggregating them.

**Minimax studies in non-parametric classification.** One of the earliest works on the minimax rate of convergence in non-parametric classification is Audibert and Tsybakov (2007). These techniques were later adopted for investigating the ability of transfer learning under a distribution shift (Kpotufe and Martinet, 2018; Cai and Wei, 2019; Maity et al., 2022). All of these works consider binary classification with 0/1 loss. In comparison, our minimax investigation differs on two fronts: we extend the settings to classification with more than two classes and general cost functions.

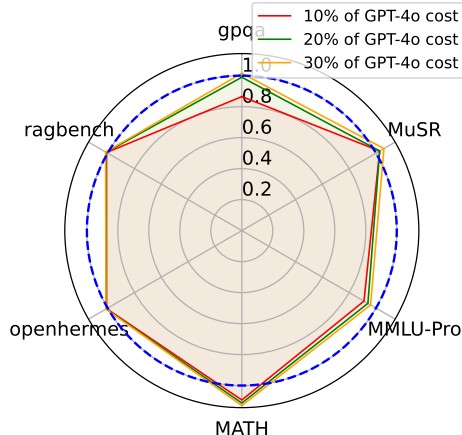

Figure 1: Percent of GPT-4o performance achieved by CARROT across datasets at various discounted costs, where the blue dotted line indicates similar (100%) performance to GPT-4o.

## 2 ROUTING PROBLEM AND PLUG-IN APPROACH

### 2.1 NOTATION AND PRELIMINARIES

To begin, let us introduce our notation. We have $M$ pre-trained LLMs indexed as $m \in [M] = \{1, \ldots, M\}$ and $K$ metrics indexed as $k \in [K] = \{1, \ldots, K\}$. We denote a generic input or query as $X \in \mathcal{X}$, where $\mathcal{X}$ is the space of inputs. Thus, for any input $X$, the metrics of interest are stored in a $M \times K$ matrix. We denote this matrix as $Y \in \mathbf{R}^{M \times K}$, whose $(m, k)$-th entry $[Y]_{m,k}$ is the metric value for obtaining a prediction from the $m$-th model evaluated with respect to $k$-th metric. For all metrics, we assume that a lower value is preferred. With this convention, we shall also refer to them as risks. For a probability distribution $P$ in the sample space $\mathcal{X} \times \mathbf{R}^{M \times K}$ we assume that the training dataset $\mathcal{D} = \{(X_i, Y_i)\}_{i=1}^{n}$ is an iid sample from $P$.

For the probability $P$ defined on the space $\mathcal{X} \times \mathbf{R}^{M \times K}$, we denote the marginal distribution of $X$ by $P_X$. Let us denote supp$(\cdot)$ as the support of a probability distribution. Within the space $\mathbf{R}^d$, we denote $\Lambda_d$ as the Lebesgue measure, $\|\cdot\|_2$ and $\|\cdot\|_\infty$ as the $\ell_2$ and $\ell_\infty$-norms, and $\mathcal{B}(x, r, \ell_2)$ and $\mathcal{B}(x, r, \ell_\infty)$ as closed balls of radius $r$ and centered at $x$ with respect to the $\ell_2$ and $\ell_\infty$-norms.

### 2.2 THE ROUTING PROBLEM

We will consider a convex combination of our $K$ metrics with coefficients $\mu \in \Delta^{K-1} \triangleq \{(\mu_1, \ldots, \mu_K) : \mu_k \geq 0, \sum_k \mu_k = 1\}$ and a generic point $(X, Y) \sim P$. The $\mu$-th convex combination of the risks (or, $\mu$-th risk) can be written as $Y\mu \in \mathbf{R}^M$, with the risk incurred for obtaining a prediction from the $m$-th model is

$$[Y\mu]_m = \sum_{k=1}^{K} [Y]_{m,k} \mu_k.$$

We want to learn a predictive router $g : \mathcal{X} \to [M]$, that takes $X$ as an input and predicts the index of the LLM to be used for inference. The average $\mu$-th risk for using the router $g$ is

$$\mathcal{R}_P(g, \mu) = \mathbf{E}\left[ \sum_{m=1}^{M} [Y\mu]_m \mathbb{I}\{g(X) = m\} \right]. \tag{2.1}$$

For a given $\mu$ let us refer to the minimizer $g_\mu^\star$ as an oracle router. The objective of the routing problem is to learn the oracle routers $g_\mu^\star$ at every value of $\mu$.

## 2.3 Plug-In Approach

While one may minimize an empirical risk corresponding to $\mathcal{R}_P(g, \mu)$ to estimate the oracle router at a particular $\mu$, this approach is not scalable, any small change in $\mu$ would require refitting a new router. Given this, we develop a plug-in approach which lets us estimate the oracle routers at every value of $\mu$. The key intuition lies within an explicit form of the $g_\mu^\star$ that we provide in the next lemma.

**Lemma 2.1.** *Let us define $\Phi(x) = \mathbf{E}[Y \mid X = x]$ and $\eta_{\mu,m}(x) = \sum_{k=1}^K \mu_k [\Phi(x)]_{m,k}$. Then for any $\mu \in \Delta^{K-1}$ the oracle router that minimizes $\mathcal{R}_P(g, \mu)$ is*

$$g_\mu^\star(X) = \arg\min_m \eta_{\mu,m}(X) = \arg\min_m \left\{ \sum_{k=1}^K \mu_k [\Phi(X)]_{m,k} \right\}.$$

The key conclusion of 2.1 is the expression $g_\mu^\star(X) = \arg\min_m \{\sum_{k=1}^K \mu_k [\Phi(X)]_{m,k}\}$. It suggests a straightforward approach to estimate $g_\mu^\star(X)$ at all values of $\mu$. Namely, we only need to plug-in an estimate of $\Phi(X) = \mathbf{E}[Y \mid X]$ to the expression of $g_\mu^\star(X)$. Compared to minimizing empirical risk at different values of $\mu$, this plug-in approach is more scalable if the practitioner plans on tuning $\mu$.

---

**CARROT**

CARROT is implemented in the following steps:

1. Learn an estimate $\widehat{\Phi}(X)$ of $\Phi(X)$ using a training split of a routing data set $\mathcal{D}_{\mathrm{tr}}$.
2. For a given convex combination of interest $\mu$, produce the router $\widehat{g}_\mu(X) = \arg\min_m \widehat{\eta}_{\mu,m}(X)$ where $\widehat{\eta}_{\mu,m}(X) = \sum_{k=1}^K \mu_k [\widehat{\Phi}(X)]_{m,k}$.

---

## 3 Statistical efficiency of CARROT

In this section we establish that, under certain conditions, the plug-in approach to routing is minimax optimal. First we establish an information theoretic lower bound on the sample complexity for learning the oracle routers (*cf.* Theorem 3.6). Next, we establish an upper bound for the minimax risk of plug-in routers (*cf.* Theorem 3.9). Finally, we show that under sufficient conditions on the estimates of $\mathbf{E}[Y \mid X]$ the sample complexity in the upper bound matches the lower bound. We will also generalize slightly to the setting where the last $K_2$ metrics are known functions of $X$, *i.e.* for $m \in [M], k \in \{K - K_2 + 1, \ldots K\}$ there exist known functions $f_{m,k} : \mathcal{X} \to \mathbf{R}$ such that $[Y]_{m,k} = f_{m,k}(X)$. Since $\mathbf{E}[[Y]_{m,k} \mid X] = f_{m,k}(X)$ are known for $k \geq K - K_2 + 1$ they don't need to be estimated.

### 3.1 Technical Assumptions

The technical assumptions of our minimax study are closely related to those in investigations of non-parametric binary classification problems with $0/1$ loss functions, *e.g.* Cai and Wei (2019); Kpotufe and Martinet (2018); Maity et al. (2022); Audibert and Tsybakov (2007). In fact, our setting generalizes the classification settings considered in these papers on multiple fronts: (i) we allow for general loss functions, (ii) we allow for more than two classes, and (iii) we allow for multiple objectives.

To clarify this, we discuss how binary classification is a special case of our routing problem.

**Example 3.1** (Binary classification with 0/1-loss)**.** *Consider a binary classification setting with 0/1-loss: we have the pairs $(X, Z) \in \mathcal{X} \times \{0, 1\}$ and we want to learn a classifier $h : \mathcal{X} \to \{0, 1\}$ to predict $Z$ using $X$. This is a special case of our setting with $M = 2$ and $K = 1$, where for $m \in \{0, 1\}$ the $[Y]_{m,1} = \mathbb{I}\{Z \neq m\}$. Then the risk for the classifier $h$, which can also be thought of as a router, is*

$$\mathcal{R}_P(h) = \mathbf{E}\left[ \sum_{m \in \{0,1\}} [Y]_{m,1} \mathbb{I}\{h(X) = m\} \right] = \mathbf{E}\left[ \mathbb{I}\{h(X) \neq Z\} \right],$$

*the standard misclassification risk for binary classification.*

We assume that $\text{supp}(P_X)$ is a compact set in $\mathbf{R}^d$. This is a standard assumption in minimax investigations for non-parametric classification problems (Audibert and Tsybakov, 2007; Cai and Wei, 2019; Kpotufe and Martinet, 2018; Maity et al., 2022). Next, we place Hölder smoothness conditions on the functions $\Phi_m^\star$. This controls the difficulty of their estimation. For a tuple $s = (s_1, \ldots, s_d) \in (\mathbf{N} \cup \{0\})^d$ of $d$ non-negative integers define $|s| = \sum_{j=1}^d s_j$ and for a function $\phi : \mathbf{R}^d \to \mathbf{R}$ and $x = (x_1, \ldots, x_d) \in \mathbf{R}^d$ define the differential operator, assuming that such a derivative exists:

$$D_s(\phi, x) = \frac{\partial^{|s|} \phi(x)}{\partial x_1^{s_1} \ldots \partial x_d^{s_d}} . \tag{3.1}$$

Using this differential operator we now define the Hölder smoothness condition:

**Definition 3.2** (Hölder smoothness). *For $\beta, K_\beta > 0$ we say that $\phi : \mathbf{R}^d \to \mathbf{R}$ is $(\beta, K_\beta)$-Hölder smooth on a set $A \subset \mathbf{R}^d$ if it is $\lfloor \beta \rfloor$-times continuously differentiable on $A$ and for any $x, y \in A$*

$$|\phi(y) - \phi_x^{(\lfloor \beta \rfloor)}(y)| \leq K_\beta \|x - y\|_2^\beta , \tag{3.2}$$

*where $\phi_x^{(\lfloor \beta \rfloor)}(y) = \sum_{|s| \leq \lfloor \beta \rfloor} D_s(\phi, x) \{\prod_{j=1}^d (y_j - x_j)^{s_j}\}$ is the $\lfloor \beta \rfloor$-order Taylor polynomial approximation of $\phi(y)$ around $x$.*

With this definition, we assume the following:

**Assumption 3.3.** *For $m \in [M]$ and $k \in [K_1]$ the $[\Phi(X)]_{m,k}$ is $(\gamma_k, K_{\gamma,k})$-Hölder smooth.*

This smoothness parameter will appear in the sample complexity of our plug-in router. Since the $[\Phi(X)]_{m,k}$ are known for $k \geq K_1 + 1$ we do not require any smoothness assumptions on them.

Next, we introduce the *margin condition*, which quantifies the difficulty in learning the oracle router. For a given $\mu$ define the margin as the difference between the minimum and second minimum of the risk values:

$$\Delta_\mu(x) = \begin{cases} \min_{m \notin g_\mu(x)} \eta_{\mu,m}(x) - \min_m \eta_{\mu,m}(x) & \text{if } g_\mu^\star(x) \neq [M] \\ 0 & \text{otherwise.} \end{cases} \tag{3.3}$$

The margin determines the difficulty in learning the oracle router. A query $X$ with a small margin gap is difficult to route, because to have the same prediction as the oracle, *i.e.* $\arg\min_m \hat{\eta}_{\mu,m}(X) = \arg\min_m \eta_{\mu,m}^\star(X)$ we need to estimate $\eta_{\mu,m}^\star(X)$ with high precision. In the following assumption, we control the probability of drawing these "difficult to route" queries.

**Assumption 3.4** (Margin condition). *For $\alpha, K_\alpha > 0$ and any $t > 0$ the margin $\Delta_\mu$ (3.3) satisfies:*

$$P_X\{0 < \Delta_\mu(X) \leq t\} \leq K_\alpha t^\alpha . \tag{3.4}$$

Following Audibert and Tsybakov (2007), we focus on the cases where $\alpha < d$ and for every $k$ the $\alpha\gamma_k < d$. This helps to avoid trivial cases where routing decisions are constant over $P_X$ for some $\mu$. Next, we assume that $P_X$ has a density $p_X$ that satisfies a strong density condition described below.

**Assumption 3.5** (Strong density condition). *Fix constants $c_0, r_0 > 0$ and $0 \leq \mu_{\min} \leq \mu_{\max} < \infty$. We say $P_X$ satisfies the strong density condition if its support is a compact $(c_0, r_0)$-regular set and it has density $p_X$ which is bounded: $\mu_{\min} \leq p_X(x) \leq \mu_{\max}$ for all $x$ within $\text{supp}(P_X)$. A set $A \subset \mathbf{R}^d$ is $(c_0, r_0)$-regular if it is Lebesgue measurable and for any $0 < r \leq r_0$, $x \in A$ it satisfies*

$$\Lambda_d(A \cap \mathcal{B}(x, r, \ell_2)) \geq c_0 \Lambda_d(\mathcal{B}(x, r, \ell_2)). \tag{3.5}$$

This is another standard assumption required for minimax rate studies in nonparametric classification problems (Audibert and Tsybakov, 2007; Cai and Wei, 2019). All together, we define $\mathcal{P}$, as the class of probabilities $P$ defined on the space $\mathcal{X} \times \mathcal{Y}$ for which $P_X$ is compactly supported and satisfies the strong density assumption 3.5 with parameters $(c_0, r_0, \mu_{\min}, \mu_{\max})$, and the Hölder smoothness assumption 3.3 and the $(\alpha, K_\alpha)$-margin condition in Assumption 3.4 hold. We shall establish our minimax rate of convergence within this probability class.

## 3.2 THE LOWER BOUND

Rather than the actual risk $\mathcal{R}_P(\mu, g)$, we establish a lower bound on the excess risk:

$$\mathcal{E}_P(\mu, g) = \mathcal{R}_P(\mu, g) - \mathcal{R}_P(\mu, g_\mu^\star), \tag{3.6}$$

that compares the risk of a proposed router to the oracle one. We denote $\Gamma = \{g : \mathcal{X} \to [M]\}$ as the class of all routers. For an $n \in \mathbf{N}$ we refer to the map $A_n : \mathcal{Z}^n \to \Gamma$, which takes the dataset $\mathcal{D}_n$ as an input and produces a router $A_n(\mathcal{D}_n) : \mathcal{X} \to [M]$, as an algorithm. Finally, call the class of all algorithms that operate on $\mathcal{D}_n$ as $\mathcal{A}_n$. The following Theorem describes a lower bound on the minimax risk for any such algorithm $A_n$.

**Theorem 3.6.** *For an $n \geq 1$ and $A_n \in \mathcal{A}_n$ define $\mathcal{E}_P(\mu, A_n) = \mathbf{E}_{\mathcal{D}_n}\big[\mathcal{E}_P\big(\mu, A_n(\mathcal{D}_n)\big)\big]$ as the excess risk of an algorithm $A_n$. There exists a constant $c > 0$ that is independent of both $n$ and $\mu$ such that for any $n \geq 1$ and $\mu \in \Delta^{K-1}$ we have the lower bound*

$$\min_{A_n \in \mathcal{A}_n} \max_{P \in \mathcal{P}} \ \mathcal{E}_P(\mu, A_n) \geq c\big\{ \textstyle\sum_{k=1}^{K_1} \mu_k n^{-\frac{\gamma_k}{2\gamma_k + d}} \big\}^{1+\alpha}. \tag{3.7}$$

This result is a generalization of that in Audibert and Tsybakov (2007), which considers binary classification.

**Remark 3.7.** *Consider the binary classification in Example 3.1. Since $K = 1$, the lower bound simplifies to $\mathcal{O}(n^{-\gamma_1(1+\alpha)/2\gamma_1+d})$, which matches with the rate in Audibert and Tsybakov (2007, Theorem 3.5). Beyond 0/1 loss, our lower bound also establishes that the rate remains identical for other classification loss functions as well.*

## 3.3 THE UPPER BOUND

Next, we show that if the algorithm $A_n$ corresponds to CARROT, the performance of $\hat{g}_\mu$ matches the lower bound in Theorem 3.6 (*cf.* equation 3.7). En-route to attaining $\hat{g}_\mu$, we need an estimate $\widehat{\Phi}(X)$ of $\Phi(X) = \mathbf{E}_P[Y \mid X]$. We begin with an assumption for a rate of convergence for $[\widehat{\Phi}(X)]_{m,k}$.

**Assumption 3.8.** *For some constants $\rho_1, \rho_2 > 0$ and any $n \geq 1$ and $t > 0$ and almost all $X$ with respect to the distribution $P_X$ we have the following concentration bound:*

$$\max_{P \in \mathcal{P}} P\big\{ \max_{m,k} a_{k,n}^{-1} \big| [\widehat{\Phi}(X)]_{m,k} - [\Phi(X)]_{m,k} \big| \geq t \big\} \leq \rho_1 \exp\big( -\rho_2 t^2 \big), \tag{3.8}$$

*where for each $k$ the $\{a_{k,n}; n \geq 1\} \subset (0, \infty)$ is a sequence that decreases to zero.*

Using this high-level assumption, in the next theorem, we establish an upper bound on the minimax excess risk for CARROT that depends on both $a_{k,n}$ and $\mu$.

**Theorem 3.9** (Upper bound). *Assume 3.8. If all the $P \in \mathcal{P}$ satisfy the margin condition 3.4 with the parameters $(\alpha, K_\alpha)$ then there exists a $K > 0$ such that for any $n \geq 1$ and $\mu \in \Delta^{K-1}$ the excess risk for the router $\widehat{g}_\mu$ in Algorithm 2.3 is upper bounded as*

$$\max_{P \in \mathcal{P}} \mathbf{E}_{\mathcal{D}_n}\big[\mathcal{E}_P(\widehat{g}_\lambda, \lambda)\big] \leq K\big\{ \textstyle\sum_{k=1}^{K_1} \mu_k a_{k,n} \big\}^{1+\alpha}. \tag{3.9}$$

**Remark 3.10** (Rate efficient routers). *When $a_{k,n} = n^{-\gamma_k/(2\gamma_k+d)}$ the upper bound in Theorem 3.9 has the $\mathcal{O}(\{\sum_{k=1}^{K_1} \mu_k n^{-\gamma_k/(2\gamma_k+d)}\}^{1+\alpha})$-rate, which is identical to the rate in the lower bound (cf. Theorem 3.6), suggesting that the minimax optimal rate of convergence for the routing problem is*

$$\min_{A_n \in \mathcal{A}_n} \max_{P \in \mathcal{P}} \ \mathcal{E}_P(A_n, \lambda) \asymp \mathcal{O}\big( \big\{ \textstyle\sum_{k=1}^{K_1} \mu_k n^{-\frac{\gamma_k}{2\gamma_k+d}} \big\}^{1+\alpha} \big). \tag{3.10}$$

*Following this, we conclude: When $a_{k,n} = n^{-\gamma_k/(2\gamma_k+d)}$ the plug-in approach in Algorithm 2.3, in addition to being computationally efficient, is also minimax rate optimal.*

An example of an estimator $\widehat{\Phi}$ that meets the needed conditions for $a_{k,n} = n^{-\gamma_k/(2\gamma_k+d)}$ to hold is described in Appendix C.1.

## 4 ROUTING IN BENCHMARK CASE-STUDIES

We use CARROT (Algorithm 2.3) to perform routing on several benchmark datasets.

### 4.1 DATASETS

**RouterBench:** RouterBench (Hu et al., 2024) is a benchmark dataset for routing tasks consisting of approximately 30k prompts and responses from eleven ($M = 11$) different LLMs. The data includes prompts from 8 benchmarks covering commonsense reasoning, knowledge-based understanding, conversation, math, and coding.

**Open LLM leaderboard:** The Open LLM leaderboard v2* (Fourrier et al., 2024) is an open-source benchmarking platform that comprises responses and evaluations of a collection of LLMs on six benchmarks comprising a diverse collection of tasks.

**SPROUT:** We introduce (and evaluate CARROT on) SPROUT, a large and diverse dataset designed for training and evaluating routers. SPROUT integrates $M = 15$ state-of-the-art language models (see Table 2) and prompts from 6 benchmarks, including GPQA (Rein et al., 2023), MuSR (Sprague et al., 2024), MMLU-Pro (Wang et al., 2024), MATH (Hendrycks et al., 2021b), OpenHermes (Teknium, 2023), and RAGBench (Friel et al., 2025). Compared to existing routing benchmarks such as RouterBench, SPROUT offers several key advantages:

1. SPROUT encompasses a highly diverse set of questions, including instruction queries.

2. Unlike previous benchmarks, it does not rely on few-shot prompting and utilizes chat templates appropriate for each model, making it more representative of real-world use cases.

3. It leverages LLaMa-3.1-70b-Instruct (Grattafiori et al., 2024) to evaluate LLM responses against the ground truth, similarly to Ni et al. (2024). This is crucial for evaluating on open-ended instruction queries as well as mitigating errors associated with traditional automatic evaluation methods like exact match.

4. We provide input and output token counts for each LLM-prompt pair, enabling flexibility when conducting cost-aware analysis.

We have released the SPROUT in huggingface and will open-source a platform that allows practitioners to extend SPROUT by adding new queries and seamlessly evaluating state-of-the-art models on them. For further details, please refer to Appendix A.

### 4.2 PLUG-IN ESTIMATES

CARROT requires an estimate for the function $\Phi_m^\star(X) = \mathbf{E}_P[Y_m \mid X]$. In our benchmark tasks, $Y_m$ is 2-dimensional, consisting of model performance measured as accuracy and model cost measured in dollars. In all routing datasets, $Y_{\text{acc},m}$ is binary, and thus we can view its estimation as a binary classification problem, where our objective is to predict the probability that $m$-th model will answer the question $X$ correctly, *i.e.* $P_m(X) = P(Y_{\text{acc},m} = 1 \mid X))$ $Y_{\text{acc},m}$. We train several multi-label classification models $\widehat{P} : \mathcal{X} \to [0,1]^M$ on a training data split consisting of 80% of the full dataset, where the $m$-th coordinate of $\widehat{P}(X)$ is the predicted probability that $m$-th model accurately answers the question $X$. In the RouterBench and SPROUT task the cost must also be estimated. We train multi-label regression models $\widehat{C} : \mathcal{X} \to \mathbf{R}^M$, where $\hat{C}_m(X) = \mathbb{E}[Y_{\text{cost},m}|X]$ is the estimated cost of calling model $m$ for query $X$. To train the cost or performance predictors we consider two procedures:

1. **CARROT (KNN):** We embed the model inputs using the text-embedding-3-small model from OpenAI (OpenAI, 2023). On these text embeddings, we train a multi-label K-nearest-neighbors (KNN) classifier/regressor.

---

*https://huggingface.co/spaces/open-llm-leaderboard/open_llm_leaderboard

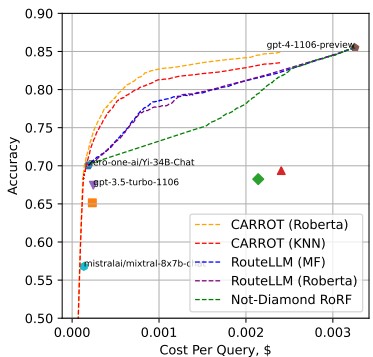 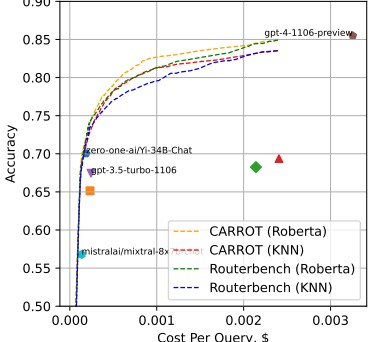

(a) CARROT vs. Binary Routers on Router-bench.

(b) CARROT vs. Routerbench on Router-bench.

Figure 2: Performance of several routers and individual LLMs on test data-split in Router-bench.

2. **CARROT (Roberta):** We fine-tune the pre-trained weights of the roberta-base[†] architecture. In order to enhance efficiency, across $m$ we allow $\widehat{P}$ to share the same network parameters, except for the final classification/regression layer.

### 4.3 BASELINE METHODS

*Zero Router:* The zero router is a simple check to see if predictive routing is of any value on a given data set. This benchmark randomly assigns prompts to the best performing (in terms of cost/accuracy trade-off) in the dataset.

*Binary Routers:* Ong et al. (2024) (RouteLLM) proposes a collection of methods for learning binary routers from preference data (data consisting of queries $q$ and labels $l_{i,j}$ indicating a winner between model $i$ and $j$). While the usage of preference data is slightly different from ours, we implement their methods on our data by creating pseudo-preference data between two models. In particular, we select a costly and high-performing model and a cheaper model and say the costly model wins if and only if it is correct while the cheaper model is incorrect. On this pseudo preference data, we fit two methods from Ong et al. (2024) for learning win probabilities between expensive and cheap models: the first is a matrix factorization method, called **RouteLLM (MF)**, while the second uses fine-tuned roberta-base, called **RouteLLM (Roberta)**. A follow-up method to these is Routing on Random Forests (RoRF) from Not-Diamond (Jain et al., 2023), referred to as **Not-Diamond RoRF**. This method uses a text-embedder and random forest model to predict the win probability; we provide a comparison to this method with the text-embedding-3-small embedder from OpenAI.

*Cost-Unaware Routers:* Another class of routers routes to multiple models but does not attempt to predict inference cost for each question (Chen et al., 2022; Nguyen et al., 2025; Hu et al., 2024). The most comparable router to ours is the **Routerbench** router proposed in (Hu et al., 2024). Their router is essentially CARROT with a constant cost predictor: For each test question, the predicted model use cost is the average cost over the training set for that model, while the performance predictor matches what we describe in Subsection 4.2

### 4.4 RESULTS

**Performance against baselines:** In Figures 2a and 3b, we see that CARROT handily beats routers that only consider two models. This is due to the fact that we route to *all possible models*, which increases the accuracy coverage and decreases the cost of the cheapest accurate model for a given query. In Figures 2b and 3a, we see that CARROT offers marginal improvements over the Routerbench router (Hu et al., 2024) that does not

---

[†]https://huggingface.co/FacebookAI/roberta-base

attempt to predict cost. Together, these findings suggest that while good cost prediction is important for achieving the most efficient price/accuracy trade-off, the large majority of routing performance is made up of correctly assessing the accuracy of each model in the family.

**CARROT can (sometimes) out-perform the best model:** In RouterBench we were unable to achieve significantly better accuracy than GPT-4; however, we were able to greatly reduce the prediction cost. Likewise on SPROUT we are able to process the test set at a fraction of the cost of o3-mini, but at its best CARROT cannot exceed the o3-mini's accuracy. On the other hand, we showed that CARROT can outperform the best model (Qwen2-72B) by a large margin in Open LLM leaderboard v2 (see Figure 3b). The difference is likely due to the existence of a singular top-performing model, or multiple models with comparable best accuracies.

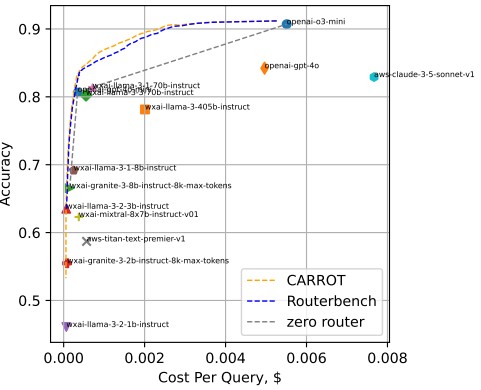

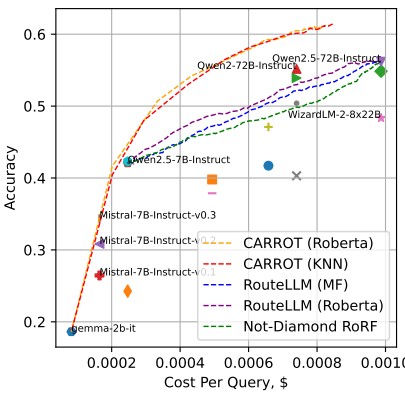

(a) CARROT vs. Routerbench and zero router on SPROUT.

(b) CARROT vs. Binary Routers on Open-LLM-Leaderboard v2.

Figure 3: CARROT routing analysis on the SPROUT and Open-LLM-Leaderboard-v2 dataset.

**Predictive routing is highly valuable for SPROUT:** Hu et al. (2024) conclude that "none of the routing algorithms significantly outperform the baseline zero router" on the Routerbench data set, where the zero router linearly interpolates between models on the frontier of cost and accuracy. This suggests that on Routerbench, predictive routing does not provide any significant benefit. In our view, this is not a shortcoming of predictive routing; rather, it is a shortcoming of the Routerbench dataset itself. Indeed, in Figure 3a we show that for a carefully collected dataset, predictive routing can provide substantial gains. On SPROUT both CARROT and the cost-unaware router provide substantial improvement over the zero-router. This suggests that SPROUT is an important introduction to the landscape of routing data sets.

## 5  DISCUSSION

We introduced CARROT, a plug-in based router that is both computationally and statistically efficient. The computational efficiency stems from the requirement of merely calculating the plug-in estimators (see Algorithm 2.3) to perform routing Since collecting adequate data for router training might be challenging, we investigate CARROT's statistical efficiency in routing through a minimax rate study. To establish the statistical efficiency of CARROT, we have provided an information-theoretic lower bound on the excess risk of any router in Theorem 3.6 and corresponding upper bound for CARROT in Theorem 3.9. To ensure a broad scope for CARROT to a diverse set of queries *and* the latest state-of-the-art LLMs, we also introduced the SPROUT dataset.

## 6 REPRODUCIBILITY STATEMENT

Appendix C includes proofs of all theoretical statements. Appendices A and D include details of the construction of SPROUT. Appendix B includes details for experiments that take place on Routerbench and Open-LLM-Leaderboard-V2. Code has been uploaded into the supplementary material, and a link to the SPROUTdataset will be included in the camera ready version.

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

# A SPROUT CONSTRUCTION DETAILS AND PLOTS

In this section, we discuss data details for SPROUT. SPROUT will be released on Hugging-Face hub as a HuggingFace datasets object. For convenience, the data is pre-divided into train, validation, and test splits. Consider the training set as an example; the features of this split are

```
features = ['key', 'dataset', 'dataset level', 'dataset idx', 'prompt',
            'golden answer', 'o3-mini', 'aws-claude-3-5-sonnet-v1',
            'titan-text-premier-v1', 'openai-gpt-4o',
            'openai-gpt-4o-mini', 'granite-3-2b-instruct',
            'granite-3-8b', 'llama-3-1-70b-instruct',
            'llama-3-1-8b-instruct', 'llama-3-2-1b-instruct',
            'llama-3-2-3b-instruct', 'llama-3-3-70b-instruct',
            'llama-3-405b-instruct', 'mixtral-8x7b-instruct-v01']
```

Each key corresponds to another list. `"prompt"` contains the model queries, the `"dataset"` list indicates which sub-task a given query falls in (*cf.* Table 1 for info), and `golden answer` contains a desirable response for each query. Finally, the model keys each correspond to a list of dictionaries that contains further information on the responses of that model. The important keys in each dictionary of the list are `["num input tokens", "num output tokens", "response", "score"]`. They contain the number of input tokens for a query, the number of output tokens a model gives in response to a query, the actual response of the model, and finally the score that the judge provides for the response (using the corresponding `golden answer` entry). The conversion of token count to cost is given in Table 2 and additional details on the judging process are described in Section A.2.

## A.1 SPROUT INGREDIENTS

Table 1 gives the benchmark ingredients for SPROUT. Namely, we use the MATH Lvl 1-5 (Hendrycks et al., 2021b), MMLU-PRO (Wang et al., 2024), GPQA (Rein et al., 2023), MUSR (Sprague et al., 2023), RAGBench (Friel et al., 2025), and openhermes (Teknium, 2023) datasets. These six benchmarks are varied and designed to simulate real-world scenarios where LLMs encounter a wide range of prompts. MATH focuses solely on mathematical word problems, whereas MMLU-PRO and GPQA include both mathematical and advanced science questions. MuSR serves as a benchmark for assessing multistep soft reasoning tasks framed within natural language narratives. RAGBench is a retrieval augmented generation (RAG) benchmark dataset collected from Question-Answer (QA) datasets (CovidQA (Möller et al., 2020), PubmedQA (Jin et al., 2019), HotpotQA (Yang et al., 2018), MS Marco (Nguyen et al., 2017), CUAD (Hendrycks et al., 2021a), EManual (Nandy et al., 2021), TechQA (Castelli et al., 2020), FinQA (Chen et al., 2021), TAT-QA (Zhu et al., 2021), ExpertQA (Malaviya et al., 2024), HAGRID (Kamalloo et al., 2023)), as well as one that was specifically adapted for RAG (DelucionQA (Sadat et al., 2023)). This measures the ability of a LLM to incorporate retrieved documents along with user queries to generate accurate answers for problems that require in-depth domain knowledge. As such, RAGbench is grouped by the needed domain knowledge: bio-medical research (PubmedQA, CovidQA), general knowledge (HotpotQA, MS Marco, HAGRID, ExperQA), legal contracts (CuAD), customer support (DelucionQA, EManual, TechQA), and finance (FinBench, TAT-QA). Finally, openhermes is a collection of GPT4 generated questions designed to emulate real user queries to an LLM.

## A.2 SPROUT MODELS AND RESPONSE COLLECTION

Table 2 provides the models and their associated costs that a router trained on SPROUT can select between. The input and output token counts are collected by simply gathering the count of the tokenized queries and outputs of a model from its tokenizer. In order to emulate real-world use cases, responses from each LLM are collected using a corresponding *chat template* with a generic prompt and *zero shot prompting*.

Table 1: Dataset Splits for SPROUT.

| Benchmark | Train | Validation | Test |
|---|---|---|---|
| ragbench/expertqa | 98 | 17 | 16 |
| MATH (test) | 1725 | 363 | 384 |
| ragbench (emanual) | 82 | 27 | 23 |
| ragbench (cuad) | 151 | 35 | 29 |
| MuSR | 178 | 35 | 35 |
| MATH | 5217 | 1061 | 1134 |
| MuSR (team allocation) | 157 | 52 | 41 |
| ragbench (hagrid) | 92 | 23 | 17 |
| gpqa (extended) | 368 | 89 | 84 |
| MuSR (object placements) | 169 | 47 | 34 |
| ragbench (pubmedqa) | 92 | 14 | 26 |
| ragbench (hotpotqa) | 89 | 22 | 21 |
| ragbench (msmarco) | 85 | 24 | 23 |
| ragbench (techqa) | 85 | 24 | 23 |
| MMLU-Pro | 8204 | 1784 | 1798 |
| openhermes | 13703 | 2917 | 2835 |
| ragbench (tatqa) | 90 | 17 | 25 |
| ragbench (finqa) | 97 | 15 | 20 |
| ragbench (covidqa) | 162 | 38 | 41 |
| ragbench (delucionqa) | 124 | 32 | 28 |
| TOTAL | 30968 | 6636 | 6637 |

Table 2: Models in SPROUT dataset and their API prices according to token counts.

| Model | Input Token Cost (in $ per 1M tokens) | Output Token Cost (in $ per 1M tokens) |
|---|---|---|
| openai-o3-mini | 1.1 | 4.4 |
| claude-3-5-sonnet-v1 | 3 | 15 |
| titan-text-premier-v1 | 0.5 | 1.5 |
| openai-gpt-4o | 2.5 | 10 |
| openai-gpt-4o-mini | 0.15 | 0.6 |
| openai-o3-mini | 1.1 | 4.4 |
| granite-3-2b-instruct | 0.1 | 0.1 |
| granite-3-8b-instruct | 0.2 | 0.2 |
| llama-3-1-70b-instruct | 0.9 | 0.9 |
| llama-3-1-8b-instruct | 0.2 | 0.2 |
| llama-3-2-1b-instruct | 0.06 | 0.06 |
| llama-3-2-3b-instruct | 0.06 | 0.06 |
| llama-3-3-70b-instruct | 0.9 | 0.9 |
| mixtral-8x7b-instruct | 0.6 | 0.6 |
| llama-3-405b-instruct | 3.5 | 3.5 |

Given the use of chat templates and zero-shot prompting, evaluation is challenging because model responses will not necessarily follow a specific format. To alleviate this, we adopt the evaluation protocol from MixEval (Ni et al., 2024) and use LLama-3.1-70B as a grader to score model queries against a given gold standard answer. The prompt format that we use is provided in D. Note that this prompt format needs to be converted to openai-api compatible messages while prompting the LLMs, which can be inferred from the special delimiters contained within the prompt format.

# B  ADDITIONAL PLOTS AND EXPERIMENTAL DETAILS

## B.1  ROUTERBENCH

Figure 4 lays out the models and benchmarks present in the Routerbench dataset. To implement the transformer-based plug-in estimate of cost and accuracy, we utilize the `roberta-base` architecture with a learning rate of `3e-5` and a weight decay of 0.01. A training, validation, test split of 0.72, 0.8, 0.2 is used. Learning proceeds for 5 epochs, and the model with the best validation performance is saved at the end. To fit the KNN-based router, the OpenAI text-embedding-small-3 model is used, while the KNN regressor utilizes the 40-nearest neighbors measured by the 'cosine' similarity metric.

The same `roberta-base` parameters are used to fit the Roberta technique from RouteLLM (Ong et al., 2024). The matrix factorization method assumes that

$$\mathbf{P}(\text{GPT-4 Win}|q) = \sigma(w_2^T(v_{\text{GPT-4}} \odot (W_1^T v_q + b) - v_{\text{mixtral}} \odot (W_1^T v_q + b)))$$

where $v_{\text{GPT-4}}$, $v_{\text{mixtral}}$ are learnable embeddings of the model of interest. We use the `text-embeddder-small-3` from OpenAI to embed the queries, and a projection dimension of $d = 128$. The model is fit using Adam, with a learning rate of $3e - 4$ and a weight decay of $1e - 5$. To fit RoRF from not-diamond, we again use `text-embeddder-small-3` while the default parameters from Not-Diamond are used (max-depth $= 20$, 100 estimators).

| Method | MMLU Perf↑ | MMLU Cost↓ | MT-Bench Perf↑ | MT-Bench Cost↓ | MBPP Perf↑ | MBPP Cost↓ | HellaSwag Perf↑ | HellaSwag Cost↓ | Winogrande Perf↑ | Winogrande Cost↓ | GSM8k Perf↑ | GSM8k Cost↓ | ARC Perf↑ | ARC Cost↓ |
|---|---|---|---|---|---|---|---|---|---|---|---|---|---|---|
| WizardLM 13B | 0.568 | 0.122 | 0.796 | 0.006 | 0.364 | 0.011 | 0.636 | 0.727 | 0.512 | 0.040 | 0.510 | 0.354 | 0.660 | 0.068 |
| Mistral 7B | 0.562 | 0.081 | 0.779 | 0.003 | 0.349 | 0.006 | 0.541 | 0.485 | 0.562 | 0.027 | 0.409 | 0.210 | 0.642 | 0.046 |
| Mixtral 8x7B | 0.733 | 0.245 | 0.921 | 0.012 | 0.573 | 0.023 | 0.707 | 1.455 | 0.677 | 0.081 | 0.515 | 0.594 | 0.844 | 0.137 |
| Code Llama 34B | 0.569 | 0.317 | 0.796 | 0.015 | 0.465 | 0.021 | 0.525 | 1.882 | 0.617 | 0.104 | 0.462 | 0.752 | 0.644 | 0.177 |
| Yi 34B | 0.743 | 0.326 | 0.938 | 0.018 | 0.333 | 0.031 | 0.931 | 1.938 | 0.748 | 0.107 | 0.552 | 0.867 | 0.882 | 0.182 |
| GPT-3.5 | 0.720 | 0.408 | 0.908 | 0.026 | 0.651 | 0.044 | 0.816 | 2.426 | 0.630 | 0.134 | 0.601 | 1.170 | 0.855 | 0.228 |
| Claude Instant V1 | 0.384 | 0.327 | 0.863 | 0.030 | 0.550 | 0.064 | 0.801 | 1.943 | 0.512 | 0.108 | 0.626 | 1.300 | 0.821 | 0.183 |
| Llama 70B | 0.647 | 0.367 | 0.854 | 0.022 | 0.302 | 0.039 | 0.736 | 2.183 | 0.504 | 0.121 | 0.529 | 0.870 | 0.794 | 0.205 |
| Claude V1 | 0.475 | 3.269 | 0.938 | 0.361 | 0.527 | 0.607 | 0.841 | 19.43 | 0.570 | 1.077 | 0.653 | 11.09 | 0.889 | 1.829 |
| Claude V2 | 0.619 | 3.270 | 0.854 | 0.277 | 0.605 | 0.770 | 0.421 | 19.50 | 0.446 | 1.081 | 0.664 | 13.49 | 0.546 | 1.833 |
| GPT-4 | 0.828 | 4.086 | 0.971 | 0.721 | 0.682 | 1.235 | 0.923 | 24.29 | 0.858 | 1.346 | 0.654 | 19.08 | 0.921 | 2.286 |
| *Oracle* | **0.957** | 0.297 | **0.996** | 0.052 | **0.899** | 0.041 | **0.994** | 0.860 | **1.0** | 0.042 | **0.748** | 1.282 | **0.977** | 0.091 |

Figure 4: Routerbench models and benchmarks (Hu et al. (2024) Table 1).

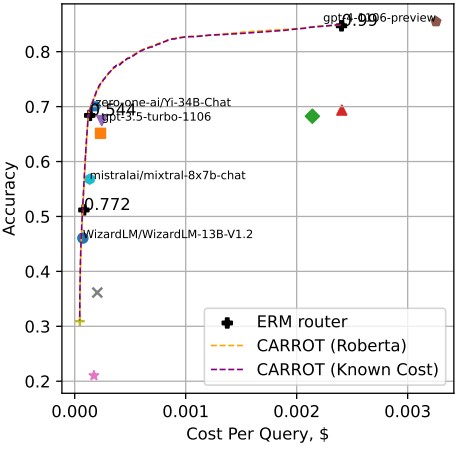

Figure 5: Router Bench Supplementary.

## B.2  OPEN LLM LEADERBOARD V2

**LLMs and costs:**  Table 3 gives all models used for the Open LLM Leaderboard experiment and their respective costs.

Table 3: Models used and their respective costs for the Open LLM Leaderboard experiment.

| Model Name | Price (USD per 1M tokens) |
|---|---|
| NousResearch/Nous-Hermes-2-Mixtral-8x7B-DPO | 0.6 |
| 01-ai/Yi-34B-Chat | 0.8 |
| Qwen/QwQ-32B-Preview | 1.2 |
| Qwen/Qwen2-72B-Instruct | 0.9 |
| Qwen/Qwen2.5-7B-Instruct | 0.3 |
| Qwen/Qwen2.5-72B-Instruct | 1.2 |
| alpindale/WizardLM-2-8x22B | 1.2 |
| deepseek-ai/deepseek-llm-67b-chat | 0.9 |
| google/gemma-2-27b-it | 0.8 |
| google/gemma-2-9b-it | 0.3 |
| google/gemma-2b-it | 0.1 |
| meta-llama/Llama-2-13b-chat-hf | 0.3 |
| meta-llama/Meta-Llama-3.1-70B-Instruct | 0.9 |
| mistralai/Mistral-7B-Instruct-v0.1 | 0.2 |
| mistralai/Mistral-7B-Instruct-v0.2 | 0.2 |
| mistralai/Mistral-7B-Instruct-v0.3 | 0.2 |
| mistralai/Mixtral-8x7B-Instruct-v0.1 | 0.6 |
| nvidia/Llama-3.1-Nemotron-70B-Instruct-HF | 0.9 |

**Model fitting:** The model fitting details for baseline methods are all the same as in the RouterBench experiment (following the original implementations). To fit our methods, we employ some hyperparameter tuning for both KNN and `roberta-base`. For KNN, we employ 5-fold cross-validation using ROC-AUC and the possible number of neighbors as 2, 4, 8, 16, 32, 64, 128, 256, or 512. For `roberta-base` hyperparameter tuning, we train for 3k steps, using 20% of the training data for validation, a batch size of 8, and search for the best combination of learning rate, weight decay, and gradient accumulation steps in {5e-5, 1e-5}, {1e-2, 1e-4}, and {1, 2, 4, 8}. The final model is trained for 10k steps.

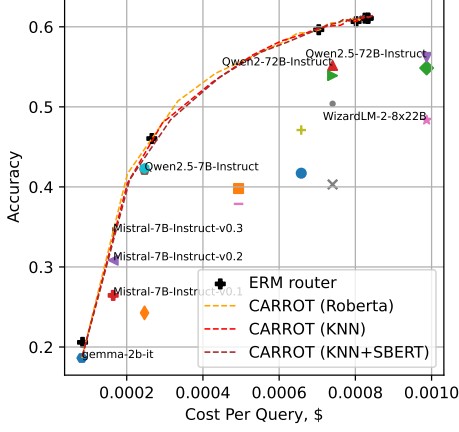

Figure 6: Open LLM leaderboard v2.

# C SUPPLEMENTARY DEFINITIONS, RESULTS AND PROOFS

## C.1 MINIMAX APPROACHES TO LEARNING THE RISK FUNCTIONS

In remark 3.10 we discussed the required condition for $\widehat{\Phi}$ so that the plug-in router has minimax rate optimal excess risk. In this section we show that estimating $\widehat{\Phi}$ using *local*

*polynomial regression* (LPR) meets the requirement. To describe the LPR estimates consider a kernel $\psi : \mathbf{R}^d \to [0, \infty)$ that satisfies the regularity conditions described in the Definition C.2 in Appendix C with parameter $\max_k \gamma_k$ and define $\Theta(p)$ as the class of all $p$-degree polynomials from $\mathbf{R}^d$ to $\mathbf{R}$. For bandwidths $h_k > 0; k \in [K_1]$ we define the LPR estimate as

$$[\widehat{\Phi}(x_0)]_{m,k} = \hat{\theta}^{(m,k)}_{x_0}(0);$$

$$\hat{\theta}^{(m,k)}_{x} \in \underset{\theta \in \Theta(p)}{\arg\min} \sum_i \psi\left(\frac{X_i - x_0}{h}\right)\left\{[Y_i]_{m,k} - \theta(X_i - x_0)\right\}^2. \tag{C.1}$$

In Theorem 3.2 of Audibert and Tsybakov (2007), a similar rate of convergence for LPR estimates is established. In their case, the losses were binary. For our instance, we assume that the $Y_i$ are sub-Gaussian, but the conclusions are identical. We restate their result below.

**Lemma C.1.** *Assume that $Y_i$ are sub-Gaussian random variables, i.e. there exist constants $c_1$ and $c_2$ such that*

$$P\left(\|Y_i\|_\infty > t \mid X\right) \le c_1 e^{-c_2 t^2}.$$

*If $\psi$ is regular (cf. Definition C.2) with parameter $\max_k \gamma_k$ and $p \ge \lfloor \max_k \gamma_k \rfloor$ then for $h_k = n^{-1/(2\gamma_k + d)}$ the Assumption 3.8 is satisfied with $a_{k,n} = n^{-\gamma_k/(2\gamma_k+d)}$, i.e. for some constants $\rho_1, \rho_2 > 0$ and any $n \ge 1$ and $t > 0$ and almost all $X$ with respect to $P_X$ we have the following concentration bound for $\widehat{\Phi}$:*

$$\max_{P \in \mathcal{P}} P\left\{ \max_{m,k} a_{k,n}^{-1} \left|[\widehat{\Phi}(X)]_{m,k} - [\Phi(X)]_{m,k}\right| \ge t \right\}$$

$$\le \rho_1 \exp\left(-\rho_2 t^2\right). \tag{C.2}$$

This result is related to our Remark 3.10 about the rate-efficient estimation of routers. Estimating $\Phi(X)$ with an LPR and a suitable bandwidth and polynomial degree leads to our desired rate of convergence $a_{k,n} = n^{-\gamma_k/(2\gamma_k+d)}$ in Assumption 3.8.

## C.2 Examples, additional assumptions and lemmas

Next, we describe the regularity conditions needed for local polynomial regression in eq. (C.1) and (C.1). These conditions are taken directly from Audibert and Tsybakov (2007, Section 3).

**Definition C.2** (Kernel regularity). *For some $\beta > 0$ we say that a kernel $K : \mathbf{R}^d \to [0, \infty)$ satisfies the regularity condition with parameter $\beta$, or simply $\beta$-regular if the following are true:*

$$\text{for some } c > 0, K(x) \ge c, \quad \text{for } \|x\|_2 \le c,$$

$$\int K(x) dx = 1$$

$$\int (1 + \|x\|_2^{4\beta}) K^2(x) dx < \infty,$$

$$\sup_x (1 + \|x\|_2^{2\beta}) K(x) < \infty.$$

An example of a kernel that satisfies these conditions is the Gaussian kernel: $K(x) = \prod_{j=1}^d \phi(x_j)$, where $\phi$ is the density of a standard normal distribution.

Next, we establish sufficient conditions for a class of distributions $\{p_\theta, \theta \in \mathbf{R}\}$ to satisfy the condition that $\mathrm{KL}(p_\theta, p_{\theta'}) \le K(\theta - \theta')^2$ for some $K > 0$ and any $\theta, \theta' \in \mathbf{R}$.

**Lemma C.3.** *Assume that a parametric family of distributions $\{p_\theta, \theta \in \mathbf{R}\}$ satisfies the following conditions:*

*1. The distributions have a density $p_\theta$ with respect to a base measure $\mu$ such that $p_\theta$ is twice continuously differentiable with respect to $\theta$.*

*2. $\int \partial_\theta p_\theta(x) d\mu(x) = \partial_\theta \int p_\theta(x) d\mu(x) = 0$*

*3. For some $K > 0$ and all $\theta \in \mathbf{R}$ the $-\partial_\theta^2 \int \log p_\theta(x) p_\theta(x) d\mu(x) \le K$.*

*Then $\mathrm{KL}(p_\theta, p_{\theta'}) \le \frac{K(\theta - \theta')^2}{2}$.*

Some prominent examples of such family are location families of normal, binomial, Poisson distributions, etc.

*Proof of the Lemma C.3.* Notice that

$$\mathrm{KL}(\mu_\theta, \mu_{\theta'})$$
$$= \int p_\theta(x) \log\left\{\frac{p_\theta(x)}{p_{\theta'}(x)}\right\} d\mu(x)$$
$$= \int p_\theta(x)\left\{\log p_\theta(x) - \log p_{\theta'}(x)\right\} d\mu(x)$$
$$= \int p_\theta(x)\left\{\log p_\theta(x) - \log p_\theta(x) - (\theta' - \theta)\partial_\theta \log p_\theta(x) - \frac{(\theta'-\theta)^2}{2}\partial_\theta^2 \log p_{\tilde\theta}(x)\right\} d\mu(x)$$

Here, $\int p_\theta(x)\partial_\theta \log p_\theta(x) d\mu(x) = \int \partial_\theta p_\theta(x) d\mu(x) dx = 0$ and $-\int p_\theta(x)\partial_\theta^2 \log p_{\tilde\theta}(x) d\mu(x) \le K$. Thus, we have the upper bound $\mathrm{KL}(\mu_\theta, \mu_{\theta'}) \le \frac{K}{2}(\theta - \theta')^2$. $\qquad\square$

## C.3 Proof of Lemma 2.1

*Proof of Lemma 2.1.* The $\mu$-th risk

$$\mathcal{R}_P(g, \mu) = \mathbf{E}\big[\mathbf{E}\big[Y\mu\big]_m \mid X\big]\mathbb{I}\{g(X) = m\}\big]$$
$$= \mathbf{E}\big[\big\{\textstyle\sum_{k=1}^K \mu_k[\Phi(X)]_{m,k}\big\}\mathbb{I}\{g(X) = m\}\big]$$

is minimized at $g(X) = \arg\min_m\big\{\sum_{k=1}^K \mu_k[\Phi(X)]_{m,k}\big\}$. $\qquad\square$

## C.4 The upper bound

**Lemma C.4.** *Suppose that we have a function $f : \mathcal{X} \to \mathbf{R}^M$ for which we define the coordinate minimizer $g : \mathcal{X} \to [M]$ as $g(x) = \arg\min_m f_m(x)$ and the margin function*

$$\Delta(x) = \begin{cases} \min_{m \ne g(x)} f_m(x) - f_{g(x)}(x) & \text{if } g(x) \ne [M] \\ 0 & \text{otherwise}. \end{cases}$$

*Assume that the margin condition is satisfied, i.e. there exist $\alpha, K_\alpha$ such that*

$$P_X\big\{0 < \Delta(X) \le t\big\} \le K_\alpha t^\alpha. \tag{C.3}$$

*Additionally, assume that there exists an estimator $\widehat{f}$ of the function $f$ such that it satisfies a concentration bound: for some $\rho_1, \rho_2 > 0$ and any $n \ge 1$ and $t > 0$ and almost all $x$ with respect to $P_X$ we have the following concentration bound for $\widehat\Phi$:*

$$P_{\mathcal{D}_n}\big\{\|\widehat{f}(x) - f(x)\|_\infty \ge t\big\} \le \rho_1 \exp\big(-\rho_2 a_n^{-2} t^2\big), \tag{C.4}$$

*where $\{a_n; n \ge 1\} \subset \mathbf{R}$ is a sequence that decreases to zero. Then for $\widehat{g}(x) = \arg\min_m \widehat{f}_m(x)$ there exists a $K > 0$ such that for any $n \ge 1$ we have the upper bound*

$$\mathbf{E}_{\mathcal{D}_n}\big[\mathbf{E}_P\big[f_{\widehat{g}(X)}(X) - f_{g(X)}(X)\big]\big] \le K a_n^{1+\alpha}. \tag{C.5}$$

*Proof.* For an $x \in \mathcal{X}$ define $\delta_m(x) = f_m(x) - f_{g(x)}(x)$. Since $g(x) = \arg\min_m f_m(x)$ we have $\delta_m(x) \ge 0$ for all $m$, $\min_m \delta_m(x) = 0$. Furthermore, define $h(x) = \arg\min\{m \ne g(x) : f_m(x)\}$, *i.e.* the coordinate of $f(x)$ where the second minimum is achieved. Clearly, $\delta_{h(x)}(x) = \Delta(x)$. With these definitions, lets break down the excess risk as:

$$\mathbf{E}_{\mathcal{D}_n}\big[\mathbf{E}_P\big[f_{\widehat{g}(X)}(X) - f_{g(X)}(X)\big]\big]$$
$$= \mathbf{E}_{\mathcal{D}_n}\big[\mathbf{E}_P\big[\textstyle\sum_{m=1}^M\{f_m(X) - f_{g(X)}(X)\}\mathbb{I}\{\widehat{g}(X) = m\}\big]\big]$$
$$= \mathbf{E}_{\mathcal{D}_n}\big[\mathbf{E}_P\big[\textstyle\sum_{m=1}^M\{f_m(X) - f_{g(X)}(X)\}\mathbb{I}\{\widehat{g}(X) = m\}\mathbb{I}\{\Delta(X) \le \tau\}\big]\big]$$
$$\quad + \textstyle\sum_{i \ge 1} \mathbf{E}_{\mathcal{D}_n}\big[\mathbf{E}_P\big[\sum_{m=1}^M\{f_m(X) - f_{g(X)}(X)\}\mathbb{I}\{\widehat{g}(X) = m\}\mathbb{I}\{\tau 2^{i-1} < \Delta(X) \le \tau 2^i\}\big]\big]$$
$$\tag{C.6}$$

where $\tau = 2\rho_2^{-1/2} a_n$. We deal with the summands one by one. First, if $\Delta(X) = 0$ then all the coordinates of $f(X)$ are identical, which further implies that $f_m(X) - f_{g(X)}(X) = 0$ for any $m$. Thus,

$$\mathbf{E}_{\mathcal{D}_n}\big[\mathbf{E}_P\big[\textstyle\sum_{m=1}^M \{f_m(X) - f_{g(X)}(X)\}\mathbb{I}\{\widehat{g}(X) = m\}\mathbb{I}\{\Delta(X) \leq \tau\}\big]\big]$$

$$= \mathbf{E}_{\mathcal{D}_n}\big[\mathbf{E}_P\big[\textstyle\sum_{m=1}^M \{f_m(X) - f_{g(X)}(X)\}\mathbb{I}\{\widehat{g}(X) = m\}\mathbb{I}\{0 < \Delta(X) \leq \tau\}\big]\big]$$

If $m = g(X)$ then the summand is zero. For the other cases, $\widehat{g}(X) = m$ if $\widehat{f}(X)$ has the minimum value at the $m$-th coordinate. This further implies $\widehat{f}_m(X) \leq \widehat{f}_{g(X)}(X)$. The only way this could happen if $|\widehat{f}_m(X) - f_m(X)| \geq \delta_m(X)/2$ or $|\widehat{f}_{g(X)}(X) - f_{g(X)}(X)| \geq \delta_m(X)/2$. Otherwise, if both are $|\widehat{f}_m(X) - f_m(X)| < \delta_m(X)/2$ and $|\widehat{f}_{g(X)}(X) - f_{g(X)}(X)| < \delta_m(X)/2$ this necessarily implies

$$\widehat{f}_{g(X)}(X) < f_{g(X)}(X) + \tfrac{\delta_m(X)}{2}$$

$$= f_m(X) - \delta_m(X) + \tfrac{\delta_m(X)}{2}$$

$$= f_m(X) - \tfrac{\delta_m(X)}{2} < \widehat{f}_m(X),$$

which means for $\widehat{f}(X)$ the minimum is not achieved at the $m$-th coordinate. Now, $|\widehat{f}_m(X) - f_m(X)| \geq \delta_m(X)/2$ or $|\widehat{f}_{g(X)}(X) - f_{g(X)}(X)| \geq \delta_m(X)/2$ implies $\|\widehat{f}(X) - f(X)\|_\infty \geq \delta_m(X)/2$. With these observations we split the expectation as

$$\mathbf{E}_{\mathcal{D}_n}\big[\mathbf{E}_P\big[\{f_m(X) - f_{g(X)}(X)\}\mathbb{I}\{\widehat{g}(X) = m\}\mathbb{I}\{0 < \Delta(X) \leq \tau\}\big]\big]$$

$$= \mathbf{E}_{\mathcal{D}_n}\big[\mathbf{E}_P\big[\{f_m(X) - f_{g(X)}(X)\}\mathbb{I}\{\widehat{g}(X) = m = g(X)\}\mathbb{I}\{0 < \Delta(X) \leq \tau\}\big]\big]$$

$$+ \mathbf{E}_{\mathcal{D}_n}\big[\mathbf{E}_P\big[\{f_m(X) - f_{g(X)}(X)\}\mathbb{I}\{\widehat{g}(X) = m \neq g(X)\}\mathbb{I}\{0 < \Delta(X) \leq \tau\}\big]\big]$$

The first part is zero, whereas the second part further simplifies as:

$$\mathbf{E}_{\mathcal{D}_n}\big[\mathbf{E}_P\big[\{f_m(X) - f_{g(X)}(X)\}\mathbb{I}\{\widehat{g}(X) = m \neq g(X)\}\mathbb{I}\{0 < \Delta(X) \leq \tau\}\big]\big]$$

$$\leq \mathbf{E}_{\mathcal{D}_n}\big[\mathbf{E}_P\big[\{f_m(X) - f_{g(X)}(X)\}\mathbb{I}\big\{\|\widehat{f}(X) - f(X)\|_\infty \geq \tfrac{\delta_m(X)}{2}\big\}\mathbb{I}\{0 < \Delta(X) \leq \tau\}\big]\big]$$

$$= \mathbf{E}_P\big[\{f_m(X) - f_{g(X)}(X)\}\mathbf{E}_{\mathcal{D}_n}\big[\mathbb{I}\big\{\|\widehat{f}(X) - f(X)\|_\infty \geq \tfrac{\delta_m(X)}{2}\big\}\big]\mathbb{I}\{0 < \Delta(X) \leq \tau\}\big]$$

$$= \mathbf{E}_P\big[\delta_m(X) P_{\mathcal{D}_n}\big\{\|\widehat{f}(X) - f(X)\|_\infty \geq \tfrac{\delta_m(X)}{2}\big\}\mathbb{I}\{0 < \Delta(X) \leq \tau\}\big]$$

$$\leq \mathbf{E}_P\big[\delta_m(X)\rho_1 e^{-\frac{\rho_2 a_n^{-2}\delta_m^2(X)}{4}}\mathbb{I}\{0 < \Delta(X) \leq \tau\}\big] = \mathbf{E}_P\big[\delta_m(X)\rho_1 e^{-\frac{\delta_m^2(X)}{\tau^2}}\mathbb{I}\{0 < \Delta(X) \leq \tau\}\big]$$

Notice that $\delta_m(X) \geq \Delta(X)$ whenever $\Delta(X) > 0$. Thus, we perform a maximization on $\delta_m(X)e^{-\frac{\delta_m^2(X)}{\tau^2}}$ on the feasible set $\delta_m(X) \geq \Delta(X)$. Here, we use the result:

$$\max_{x \geq y} x e^{-\frac{x^2}{\tau^2}} \leq \begin{cases} \frac{\tau}{\sqrt{2e}} & \text{if } \frac{\tau}{\sqrt{2}} \geq y \\ y e^{-\frac{y^2}{\tau^2}} & \text{otherwise}, \end{cases} \tag{C.7}$$

where $x = \delta_m(X)$ and $y = \Delta(X)$. Since $\Delta(X) \leq \tau$ we have $\delta_m(X)e^{-\frac{\delta_m^2(X)}{\tau^2}} \leq \tau$ and thus

$$\mathbf{E}_P\big[\delta_m(X)\rho_1 e^{-\frac{\delta_m^2(X)}{\tau^2}}\mathbb{I}\{0 < \Delta(X) \leq \tau\}\big] \leq \rho_1 \tau P\{0 < \Delta(X) \leq \tau\} = \rho_1 \tau^{1+\alpha}.$$

This finally results in

$$\mathbf{E}_{\mathcal{D}_n}\big[\mathbf{E}_P\big[\textstyle\sum_{m=1}^M \{f_m(X) - f_{g(X)}(X)\}\mathbb{I}\{\widehat{g}(X) = m\}\mathbb{I}\{\Delta(X) \leq \tau\}\big]\big] \leq M\rho_1 \tau^{1+\alpha},$$

which takes care of the first summand in eq. (C.6). Now, for an $i \geq 1$, let us consider the summand

$$\mathbf{E}_{\mathcal{D}_n}\big[\mathbf{E}_P\big[\textstyle\sum_{m=1}^M \{f_m(X) - f_{g(X)}(X)\}\mathbb{I}\{\widehat{g}(X) = m\}\mathbb{I}\{\tau 2^{i-1} < \Delta(X) \leq \tau 2^i\}\big]\big]$$

Again, on the event $m = g(X)$ the the summand is zero and on the other cases we have $\|\widehat{f}(X) - f(X)\|_\infty \geq \delta_m(X)/2$. Thus, we write

$$\mathbf{E}_{\mathcal{D}_n}\big[\mathbf{E}_P\big[\textstyle\sum_{m=1}^M \{f_m(X) - f_{g(X)}(X)\}\mathbb{I}\{\widehat{g}(X) = m\}\mathbb{I}\{\tau 2^{i-1} < \Delta(X) \leq \tau 2^i\}\big]\big]$$

$$\leq \textstyle\sum_{m=1}^M \mathbf{E}_{\mathcal{D}_n}\big[\mathbf{E}_P\big[\delta_m(X)\mathbb{I}\big\{\|\widehat{f}(X) - f(X)\|_\infty \geq \tfrac{\delta_m(X)}{2}\big\}\mathbb{I}\{\tau 2^{i-1} < \Delta(X) \leq \tau 2^i\}\big]\big]$$

$$\leq \textstyle\sum_{m=1}^M \mathbf{E}_P\big[\delta_m(X)\rho_1 e^{-\frac{\delta_m^2(X)}{\tau^2}}\mathbb{I}\{\tau 2^{i-1} < \Delta(X) \leq \tau 2^i\}\big]$$

Because $\Delta(X) \geq \tau 2^{i-1} > \tau/\sqrt{2}$ we again use the inequality in eq. (C.7) to obtain

$$\sum_{m=1}^{M} \mathbf{E}_P\big[\delta_m(X)\rho_1 e^{-\frac{\delta_m^2(X)}{\tau^2}}\mathbb{I}\{\tau 2^{i-1} < \Delta(X) \leq \tau 2^i\}\big]$$

$$\leq \sum_{m=1}^{M} \mathbf{E}_P\big[\Delta(X)\rho_1 e^{-\frac{\Delta^2(X)}{\tau^2}}\mathbb{I}\{\tau 2^{i-1} < \Delta(X) \leq \tau 2^i\}\big]$$

$$\leq \sum_{m=1}^{M} \tau 2^i \rho_1 e^{-\frac{\tau^2 2^{2i-2}}{\tau^2}} P\{\tau 2^{i-1} < \Delta(X) \leq \tau 2^i\}$$

$$\leq M\tau 2^i \rho_1 e^{-\frac{\tau^2 2^{2i-2}}{\tau^2}} P\{0 < \Delta(X) \leq \tau 2^i\} = M\rho_1 \tau^{1+\alpha} 2^{i(1+\alpha)} e^{-2^{2i-2}}$$

Combining all the upper bounds in (C.6) we finally obtain

$$\mathbf{E}_{\mathcal{D}_n}\big[\mathbf{E}_P\big[f_{\widehat{g}(X)}(X) - f_{g(X)}(X)\big]\big] \leq M\rho_1 \tau^{1+\alpha}\big\{1 + \sum_{i\geq 1} 2^{i(1+\alpha)} e^{-2^{2i-2}}\big\} \qquad \text{(C.8)}$$

As $\sum_{i\geq 1} 2^{i(1+\alpha)} e^{-2^{2i-2}}$ is finite we have the result. $\qquad\square$

*Proof of Theorem 3.9.* The proof of the upper bound follows directly from the lemma C.4 once we establish that for $a_n = \sum_{k=1}^{K_1} \mu_k a_{k,n}$ the following concentration holds: for constants $\rho_1, \rho_2 > 0$ and any $n \geq 1$ and $t > 0$ and almost all $X$ with respect to $P_X$ we have

$$\max_{P \in \mathcal{P}} P\big\{\max_m \big|\widehat{\eta}_{\mu,m}(X) - \eta_{\mu,m}^{\star}(X)\big| \geq t\big\} \leq \rho_1 \exp\big(-\rho_2 a_n^{-2} t^2\big). \qquad \text{(C.9)}$$

To this end, notice that

$$\max_m \big|\widehat{\eta}_{\mu,m}(X) - \eta_{\mu,m}(X)\big|$$

$$\leq \sum_{k=1}^{K} \mu_k \max_m \big|[\widehat{\Phi}(X)]_{m,k} - [\Phi(X)]_{m,k}\big|$$

$$= \sum_{k=1}^{K_1} \mu_k \max_m \big|[\widehat{\Phi}(X)]_{m,k} - [\Phi(X)]_{m,k}\big|$$

where the last equality holds because $[\widehat{\Phi}(X)]_{m,k} = [\Phi(X)]_{m,k}$ for $k \geq K_1 + 1$. Following this inequality, we have that for any $P \in \mathcal{P}$

$$P\big\{\max_m \big|\widehat{\eta}_{\mu,m}(X) - \eta_{\mu,m}(X)\big| \geq K_1 t\big\}$$

$$\leq \sum_{k=1}^{K_1} P\big\{\max_m \big|[\widehat{\Phi}(X)]_{m,k} - [\Phi(X)]_{m,k}\big| \geq \tfrac{t}{\mu_k}\big\}$$

$$\leq \sum_{k=1}^{K_1} \rho_{k,1} \exp\big(-\rho_{k,2}\mu_k^{-2} a_{k,n}^{-2} t^2\big)$$

$$\leq \rho_1 \exp\big(-\rho_2 K_1^2 \{\wedge_{k=1}^{K_1} \mu_k^{-1} a_{k,n}^{-1}\}^2 t^2\big)$$

where $\rho_1 = \frac{\max_{k \leq K_1} \rho_{k,1}}{K_1}$ and $\rho_2 = K_1^{-2} \times \{\wedge_{k \leq K_1} \rho_{k,2}\}$. Note that

$$K_1\{\wedge_{k=1}^{K_1} \mu_k^{-1} a_{k,n}^{-1}\}^{-1} = K_1 \max_{k=1}^{K_1} \mu_k a_{l,n} \geq \sum_{k \leq K_1} \mu_k a_{k,n} = a_n.$$

Thus,

$$P\big\{\max_m \big|\widehat{\eta}_{\mu,m}(X) - \eta_{\mu,m}(X)\big| \geq K_1 t\big\}$$

$$\leq \rho_1 \exp\big(-\rho_2 K_1^2 \{\wedge_{k=1}^{K_1} \mu_k^{-1} a_{k,n}^{-1}\}^2 t^2\big) \leq \rho_1 \exp\big(-\rho_2 a_n^2 t^2\big).$$

$\qquad\square$

## C.5 The lower bound

To begin, we discuss the high-level proof strategy that will achieve our lower bound. Ultimately, for every $k \leq K_1$ we shall establish that for any $\epsilon_k \in [0,1]$ and $n \geq 1$

$$\min_{A_n \in \mathcal{A}_n} \max_{P \in \mathcal{P}} \mathcal{E}_P(\mu, A_n) \geq c_k\big\{\mu_k n^{-\frac{\gamma_k}{2\gamma_k+d}}\big\}^{1+\alpha}, \qquad \text{(C.10)}$$

for some constant $c_k > 0$. Then, defining $c = \min\{c_k : k \leq K_1\}$ we have the lower bound

$$\min_{A_n \in \mathcal{A}_n} \max_{P \in \mathcal{P}} \mathcal{E}_P(\mu, A_n) \geq \max_{k \leq K_1} c_k\big\{\mu_k n^{-\frac{\gamma_k}{2\gamma_k+d}}\big\}^{1+\alpha}$$

$$\geq \max_{k \leq K_1} c\big\{\mu_k n^{-\frac{\gamma_k}{2\gamma_k+d}}\big\}^{1+\alpha}$$

$$\geq c\big\{\sum_{k \leq K_1} \frac{\mu_k n^{-\frac{\gamma_k}{2\gamma_k+d}}}{K}\big\}^{1+\alpha}$$

$$\geq cK^{-1-\alpha}\big\{\sum_{k \leq K_1} \mu_k n^{-\frac{\gamma_k}{2\gamma_k+d}}\big\}^{1+\alpha},$$

which would complete the proof.

It remains to establish (C.10) for each $k \in [K_1]$. To obtain this, we construct a finite family of probability measures $\mathcal{M}_r \subset \mathcal{P}$ (indexed by $[r]$) and study $\max_{P \in \mathcal{M}_r}$. The technical tool which allows this to be fruitful is a generalized version of Fano's lemma.

**Lemma C.5** (Generalized Fano's lemma). *Let $r \geq 2$ be an integer and let $\mathcal{M}_r \subset \mathcal{P}$ contains $r$ probability measures indexed by $\{1, \ldots, r\}$ such that for a pseudo-metric $d$ (i.e. $d(\theta, \theta') = 0$ if and only if $\theta = \theta'$) any $j \neq j'$*

$$d\big(\theta(P_j), \theta(P_{j'})\big) \geq \alpha_r, \quad and \quad KL(P_j, P_{j'}) \leq \beta_r.$$

*Then*

$$\max_j \mathbf{E}_{P_j}\big[d(\theta(P_j), \widehat{\theta})\big] \geq \frac{\alpha_r}{2}\big(1 - \frac{\beta_r + \log 2}{\log r}\big).$$

In our construction $\theta(P^\sigma) = g^\star_{\mu,\sigma}$ and $d\big(\theta(P^{\sigma_0}), \theta(P^{\sigma_1})\big) = \mathcal{E}_{P^{\sigma_0}}(g^\star_{\mu,\sigma_1}, \mu)$.

Next, we lay out the template for constructing the family $\mathcal{M}_r$. Fix a $k_0 \in [K_1]$ and define the following.

**Definition C.6.** *1. For an $h = L \times \mu_{k_0}^{\frac{1}{\gamma_{k_0}}} n^{-\frac{1}{2\gamma_{k_0}+d}}$ ($L > 0$ is a constant to be decided later) define $m = \lfloor h^{-1} \rfloor$.*

*2. Define $\mathcal{G} = \big[\{ih + \frac{h}{2} : i = 0, \ldots, m-1\}^d\big]$ as a uniform grid in $[0,1]^d$ of size $m^d$ and $\mathcal{G}_\epsilon$ as an $\epsilon$-net in $\ell_\infty$ metric, i.e. $\mathcal{G}_\epsilon = \cup_{x \in \mathcal{G}} \mathcal{B}(x, \epsilon, \ell_\infty)$, where $\mathcal{B}(x, \epsilon, \ell_\infty) = \{y \in \mathcal{X} : \|x-y\|_\infty \leq \epsilon\}$.*

*3. Define $P_X = Unif(\mathcal{G}_\epsilon)$. For such a distribution, note that $\mathrm{vol}(\mathcal{G}_\epsilon) = (m\epsilon)^d \leq (h^{-1}\epsilon)^d$, which implies that for all $x \in \mathcal{G}_\epsilon$ we have $p_X(x) = (h\epsilon^{-1})^d$. Setting $\epsilon = p_0^{-1/d} h \wedge \frac{h}{3}$ we have $p_X(x) \geq p_0$ that satisfies the strong density assumption for $P_X$.*

*4. Fix an $m_0 \leq m^d$ and consider $\mathcal{G}_0 \subset \mathcal{G}$ such that $|\mathcal{G}_0| = m_0$ and define $\mathcal{G}_1 = \mathcal{G}\backslash\mathcal{G}_0$.*

*5. For a function $\sigma : \mathcal{G}_0 \to [M]$ define*

$$\Phi^\sigma_{m,k}(x) = \begin{cases} \frac{1 - K_{\gamma,k_0}\mu_{k_0}^{-1}\epsilon^{\gamma_{k_0}}\mathbb{I}\{\sigma(y)=m\}}{2} & when \ k = k_0, \ x \in \mathcal{B}(x, \epsilon, \ell_\infty) \ for \ some \ y \in \mathcal{G}_0, \\ \frac{1}{2} & elsewhere. \end{cases}$$

$$(C.11)$$

*6. Consider a class of probability distributions $\{\mu_\theta : \theta \in \mathbf{R}\}$ defined on the same support $\mathrm{range}(\ell)$ that have mean $\theta$ and satisfy $KL(\mu_\theta, \mu_{\theta'}) \leq c(\theta - \theta')^2$ for some $c > 0$. A sufficient condition for constricting such a family of distributions can be found in Lemma C.3. Some prominent examples of such family are location families of normal, binomial, Poisson distributions, etc. Define the probability $P^\sigma([Y]_{m,k} \mid X = x) \sim \mu_{\Phi^{(\sigma)}_{m,k}(x)}$.*

The following two lemmas (along with the observation on the strong density condition) will establish that for a given $\sigma$, the distribution over $\mathcal{X}, \mathcal{Y}$ given by $P^\sigma([Y]_{m,k} \mid X = x) \times \mathrm{Unif}[\mathcal{G}_\epsilon]$ is indeed a member of the class $\mathcal{P}$.

**Lemma C.7.** *Fix a choice for $\sigma$ and let $\eta^\sigma_{\mu,m} = \sum_k \mu_k \Phi^\sigma_{k,m}(x)$, then $\eta^\sigma_{\mu,m}$ satisfies $\alpha$-margin condition.*

*Proof.* To see that $\eta^\sigma_{\mu,m}$ satisfies $\alpha$-margin condition, notice that

$$\eta^\sigma_{\mu,m}(x) = \begin{cases} \frac{1 - K_{\gamma,k_0}\epsilon^{\gamma_{k_0}}\mathbb{I}\{\sigma(y)=m\}}{2} & when \ x \in \mathcal{B}(x, \epsilon, \ell_\infty) \ for \ some \ y \in \mathcal{G}_0, \\ \frac{1}{2} & elsewhere. \end{cases}$$

Thus, for every $x \in \mathcal{B}(y, \epsilon, \ell_\infty), y \in \mathcal{G}_0$ the $\Phi^\sigma_{\mu,m}(x) = \frac{1}{2}$ for all but one $m$ and at $m = \sigma(x)$ the $\Phi^\sigma_{\mu,m}(x) = \frac{1 - K_{\gamma,k_0}\epsilon^{\gamma_{k_0}}}{2}$, leading to $\Delta^\sigma_\mu(x) = \frac{K_{\gamma,k_0}\epsilon^{\gamma_{k_0}}}{2}$ at those $x$, and at all other $x$ we

have $\Delta_\mu^\sigma(x) = 0$. This further implies $P_X(0 < \Delta_\mu^\sigma(X) \leq t) = 0$ whenever $t < \frac{K_{\gamma,k_0}\epsilon^{\gamma_{k_0}}}{2}$ and for $t \geq \frac{K_{\gamma,k_0}\epsilon^{\gamma_{k_0}}}{2}$ we have

$$P_X(0 < \Delta^\sigma(X) \leq t) = P_X\big(\Phi_m^\sigma(X) \neq \tfrac{1}{2} \text{ for some } m \in [M]\big)$$

$$\leq m_0\epsilon^d \leq K_\alpha\big(\tfrac{K_{\gamma,k_0}\epsilon^{\gamma_{k_0}}}{2}\big)^\alpha$$

whenever

$$m_0 \leq K_\alpha 2^{-\alpha} K_{\gamma,k_0}^\alpha \epsilon^{\alpha\gamma_{k_0}-d}$$

We set $m_0 = \lfloor K_\alpha 2^{-\alpha} K_{\gamma,k_0}^\alpha \epsilon^{\alpha\gamma_{k_0}-d} \rceil$ to meet the requirement. Since $d > \min_k \alpha\gamma_k$, for sufficiently small $\epsilon$ we have $m_0 \geq 8$. $\qquad\square$

**Lemma C.8.** *On the support of $P_X$ the $\Phi_{m,k}^\sigma$ are $(\gamma_k, K_{\gamma,k})$ Hölder smooth.*

*Proof.* Note that the only way $\Phi_{m,k}^\sigma(x)$ and $\Phi_{m,k}^\sigma(x')$ can be different if $\|x - x'\|_\infty \geq \frac{h}{3}$. Since $\epsilon \leq \frac{h}{3}$ for such a choice, we have

$$|\Phi_{m,k}^\sigma(x) - \Phi_{m,k}^\sigma(x')| \leq \tfrac{1}{2}K_{\gamma,k}\epsilon^\beta$$

$$\leq K_{\gamma,k}(\tfrac{h}{3})^\beta$$

$$\leq K_{\gamma,k}\|x - x'\|_\infty^\beta \leq K_{\gamma,k}\|x - x'\|_2^\beta.$$

$\qquad\square$

In order transfer the inequality in Fano's lemma to a statement on rate of convergence, we need an upper bound on $\mathrm{KL}(P^{\sigma_1}, P^{\sigma_2})$ and a lower bound on the semi-metric $\mathcal{E}_{P^{\sigma_0}}(\mu, g_{\mu,\sigma_1}^\star)$. These are established in the next two lemmas.

**Lemma C.9.** *Consider the probability distribution $P^\sigma$ for the random pair $(X, Y)$ where $X \sim P_X$ and given $X$ the $\{[Y]_{m,k}; m \in [M], k \leq K_1\}$ are all independent and distributed as $[Y]_{m,k} \mid X = x \sim \mu_{\Phi_{m,k}^\sigma(x)}$. Let $C$ be a positive constant and $\delta(\sigma_1, \sigma_2) = \sum_{y \in \mathcal{G}_0} \mathbb{I}\{\sigma_1(y) \neq \sigma_2(y)\}$ the Hamming distance between $\sigma_1$ and $\sigma_2$. Then following upper bound holds on $\mathrm{KL}(P^{\sigma_1}, P^{\sigma_2})$.*

$$\mathrm{KL}(P^{\sigma_1}, P^{\sigma_2}) \leq C\mu_{k_0}^{-2}h^{2\gamma_{k_0}+d}\delta(\sigma_1, \sigma_2)$$

*Proof.*

$$\mathrm{KL}(P^{\sigma_1}, P^{\sigma_2})$$

$$= \int dP_X(x) \sum_{m=1}^M \sum_{k=1}^K \mathrm{KL}\big(\mu_{\Phi_{m,k}^{(\sigma_1)}(x)}, \mu_{\Phi_{m,k}^{(\sigma_2)}(x)}\big)$$

$$\leq \int dP_X(x) \sum_{m=1}^M \sum_{k=1}^K c\big(\Phi_{m,k}^{(\sigma_1)}(x) - \Phi_{m,k}^{(\sigma_2)}(x)\big)^2 \qquad (\mathrm{KL}(\mu_\theta, \mu_{\theta'}) \leq c(\theta - \theta')^2)$$

$$= \sum_{y \in \mathcal{G}_0} \epsilon^d \sum_{m=1}^M \frac{cK_{\gamma,k_0}^2\epsilon^{2\gamma_{k_0}}\mu_{k_0}^{-2}}{4}\big(\mathbb{I}\{\sigma_1(y) = m\} - \mathbb{I}\{\sigma_2(y) = m\}\big)^2$$

$$\leq \frac{cK_{\gamma,k_0}^2}{4}\sum_{y \in \mathcal{G}_0}\mu_{k_0}^{-2}\epsilon^{2\gamma_{k_0}+d} \times \mathbb{I}\{\sigma_1(y) \neq \sigma_2(y)\}$$

$$\leq C\mu_{k_0}^{-2}h^{2\gamma_{k_0}+d}\delta(\sigma_1, \sigma_2) \qquad (\text{because } \epsilon \leq \tfrac{h}{3})$$

for some $C > 0$, where $\delta(\sigma_1, \sigma_2) = \sum_{y \in \mathcal{G}_0}\mathbb{I}\{\sigma_1(y) \neq \sigma_2(y)\}$ is the Hamming distances between $\sigma_1$ and $\sigma_2$. $\qquad\square$

Now, we establish a closed form for the excess risk

$$\mathcal{E}_{P^{\sigma_0}}(\mu, g_{\mu,\sigma_1}^\star) = \mathbf{E}_{P^{\sigma_0}}(\mu, g_{\mu,\sigma_1}^\star) - \mathbf{E}_{P^{\sigma_0}}(\mu, g_{\mu,\sigma_0}^\star)$$

where $g_{\mu,\sigma_0}^\star$ is the Bayes classifier for $P^{\sigma_0}$ defined as $g_{\mu,\sigma_0}^\star(x) = \arg\min_m \Phi_{\mu,m}^{\sigma_0}(x)$.

**Lemma C.10.** *Let $\delta(\sigma_0, \sigma_1)$ denote the Hamming distance between $\sigma_0$ and $\sigma_1$ as before. Then*

$$\mathcal{E}_{P^{\sigma_0}}(\mu, g_{\mu,\sigma_1}^\star) = \frac{K_{\gamma,k_0}\epsilon^{\gamma_{k_0}+d}\delta(\sigma_0, \sigma_1)}{2}$$

*Proof.* For the purpose, notice that

$$g^\star_{\mu,\sigma}(x) = \sigma(y) \quad \text{whenever } x \in \mathcal{B}(x,\epsilon,\ell_\infty) \text{ for some } y \in \mathcal{G}_0.$$

This further implies

$$\mathbf{E}_{P^{\sigma_0}}(\mu, g^\star_{\mu,\sigma_1})$$

$$= \int dP_X(x) \sum_{m=1}^M \mathbb{I}\{g^\star_{\mu,\sigma_1}(x) = m\} \Phi^{\sigma_0}_{\mu,m}(x)$$

$$= \sum_{y \in \mathcal{G}_0} \epsilon^d \sum_{m=1}^M \mathbb{I}\{\sigma_1(y) = m\} \mu_{k_0} \tfrac{1}{2}\{1 - K_{\gamma,k_0} \mu_{k_0}^{-1} \epsilon^{\gamma_{k_0}} \mathbb{I}\{\sigma_0(y) = m\}\}$$

$$\quad + \sum_{y \in \mathcal{G}_0} \epsilon^d \sum_{m=1}^M \mathbb{I}\{\sigma_1(y) = m\} \sum_{k \neq k_0} \tfrac{\mu_k}{2} + \sum_{y \in \mathcal{G}_1} \epsilon^d \sum_{m=1}^M \mathbb{I}\{\sigma_1(y) = m\} \tfrac{1}{2}$$

$$= -\sum_{y \in \mathcal{G}_0} \sum_{m=1}^M \frac{K_{\gamma,k_0} \epsilon^{\gamma_{k_0}+d}}{2} \mathbb{I}\{\sigma_0(y) = \sigma_1(y) = m\}$$

$$\quad + \sum_{y \in \mathcal{G}_0 \cup \mathcal{G}_1} \epsilon^d \sum_{m=1}^M \mathbb{I}\{\sigma_1(y) = m\} \tfrac{1}{2}$$

$$= -\sum_{y \in \mathcal{G}_0} \sum_{m=1}^M \frac{K_{\gamma,k_0} \epsilon^{\gamma_{k_0}+d}}{2} \mathbb{I}\{\sigma_0(y) = \sigma_1(y) = m\} + \sum_{y \in \mathcal{G}_0 \cup \mathcal{G}_1} \frac{\epsilon^d}{2}$$

By replacing $\sigma_1$ with $\sigma_0$ in the above calculations we obtain

$$\mathbf{E}_{P^{\sigma_0}}(\mu, g^\star_{\mu,\sigma_0}) = -\sum_{y \in \mathcal{G}_0} \sum_{m=1}^M \frac{K_{\gamma,k_0} \epsilon^{\gamma_{k_0}+d}}{2} \mathbb{I}\{\sigma_0(y) = m\} + \sum_{y \in \mathcal{G}_0 \cup \mathcal{G}_1} \frac{\epsilon^d}{2}$$

and hence

$$\mathcal{E}_{P^{\sigma_0}}(g^\star_{\mu,\sigma_1}, \mu)$$

$$= \mathbf{E}_{P^{\sigma_0}}(g^\star_{\mu,\sigma_1}, \mu) - \mathbf{E}_{P^{\sigma_0}}(g^\star_{\mu,\sigma_0}, \mu)$$

$$= \sum_{y \in \mathcal{G}_0} \sum_{m=1}^M \frac{K_{\gamma,k_0} \epsilon^{\gamma_{k_0}+d}}{2} \{\mathbb{I}\{\sigma_0(y) = m\} - \mathbb{I}\{\sigma_0(y) = \sigma_1(y) = m\}\}$$

$$= \frac{K_{\gamma,k_0} \epsilon^{\gamma_{k_0}+d}}{2} \sum_{y \in \mathcal{G}_0} \sum_{m=1}^M \mathbb{I}\{\sigma_0(y) = m\} \times \mathbb{I}\{\sigma_1(y) \neq m\}$$

$$= \frac{K_{\gamma,k_0} \epsilon^{\gamma_{k_0}+d}}{2} \sum_{y \in \mathcal{G}_0} \mathbb{I}\{\sigma_0(y) \neq \sigma_1(y)\}$$

$$= \frac{K_{\gamma,k_0} \epsilon^{\gamma_{k_0}+d} \delta(\sigma_0,\sigma_1)}{2}.$$

$\square$

The final technical ingredient we require is the Gilbert–Varshamov bound for linear codes.

**Lemma C.11** (Gilbert–Varshamov bound). *Consider the maximal $A_M(m_0, d) \subset [M]^{m_0}$ such that each element in $C$ is at least $d$ Hamming distance from each other, i.e. for any $\sigma_1, \sigma_2 \in C$ we have $\delta(\sigma_1, \sigma_2) \geq d$. Then*

$$|A_M(m_0, d)| \geq \frac{M^{m_0}}{\sum_{i=0}^{d-1} \binom{m_0}{i}(M-1)^i}$$

*Furthermore, when $M \geq 2$ and $0 \leq p \leq 1 - \frac{1}{M}$ we have $|A_M(m_0, pm_0)| \geq M^{m_0(1-h_M(p))}$ where $h_M(p) = \frac{p \log(M-1) - p \log p - (1-p) \log(1-p)}{\log M}$.*

*Proof of the Theorem 3.6.* For the choice $p = \frac{1}{4}$ we have $-p \log p - (1-p) \log(1-p) \leq \frac{1}{4}$ and thus

$$h_M(p) \leq \frac{\log(M-1)}{4 \log M} + \frac{1}{4 \log M} \leq \frac{1}{4} + \frac{1}{4 \log 2} \leq \frac{3}{4}.$$

Consequently, the lemma implies that we can find an $A_M(m_0, \frac{m_0}{4}) \subset [M]^{m_0}$ such that $|A_M(m_0, \frac{m_0}{4})| \geq M^{\frac{m_0}{4}}$ whose each element is at least $\frac{m_0}{4}$ Hamming distance apart. For such a choice, define the collection of probabilities as $\mathcal{M}_r = \{P^\sigma : \sigma \in A_M(m_0, \frac{m_0}{4})\}$ leading to $r \geq M^{\frac{m_0}{4}}$. In the generalized Fano's lemma C.5 we require $r \geq 2$. To achieve that we simply set $m_0 \geq 8$, as it implies $r \geq M^2 \geq 4$.

Now we find lower bound $\alpha_r$ for the semi-metric and upper bound $\beta_r$ for the Kulback-Leibler divergence. Let's start with the upper bound. Since $\mathrm{KL}(P^{\sigma_1}, P^{\sigma_2}) \leq C \mu_{k_0}^{-2} h^{2\gamma_{k_0}+d} \delta(\sigma_1, \sigma_2)$

for the joint distributions of the dataset $\mathcal{D}_n$ the Kulback-Leibler divergence between $\{P^{\sigma_1}\}^{\otimes n}$ and $\{P^{\sigma_2}\}^{\otimes n}$ is upper bounded as:

$$
\begin{aligned}
&\mathrm{KL}\big(\{P^{\sigma_1}\}^{\otimes n}, \{P^{\sigma_2}\}^{\otimes n}\big) \\
&= n\mathrm{KL}(P^{\sigma_1}, P^{\sigma_2}) \\
&\leq nC\mu_{k_0}^{-2} h^{2\gamma_{k_0}+d} \delta(\sigma_1, \sigma_2) \\
&= nC\mu_{k_0}^{-2} L^{2\gamma_{k_0}+d} \mu_{k_0}^{\frac{2\gamma_{k_0}+d}{\gamma_{k_0}}} n^{-\frac{2\gamma_{k_0}+d}{2\gamma_{k_0}+d}} \qquad \text{(because } h \text{ is defined as } L \times \mu_{k_0}^{\frac{1}{\gamma_{k_0}}} n^{-\frac{1}{2\gamma_{k_0}+d}}) \\
&\leq CL^{2\gamma_{k_0}+d} \mu_{k_0}^{\frac{d}{\gamma_{k_0}}} \frac{\log r}{\log M} \qquad \text{(because } r \geq M^{\frac{m_0}{4}}) \\
&\leq CL^{2\gamma_{k_0}+d} \frac{\log r}{\log M} = \beta_r
\end{aligned}
$$

In the Lemma C.5 we would like $\frac{\beta_r+\log 2}{\log r} \leq \frac{3}{4}$ so that we have $1 - \frac{\beta_r+\log 2}{\log r} \geq \frac{1}{4}$. Note that,

$$
\begin{aligned}
\frac{\beta_r+\log 2}{\log r} - \frac{3}{4} &= \frac{\beta_r}{\log r} + \frac{\log 2}{\log r} - \frac{3}{4} \\
&= \frac{CL^{2\gamma_{k_0}+d}}{\log M} + \frac{\log 2}{\log 4} - \frac{3}{4} \qquad \text{(because } r \geq 4, \ \beta_r = CL^{2\gamma_{k_0}+d} \frac{\log r}{\log M}) \\
&= \frac{CL^{2\gamma_{k_0}+d}}{\log M} - \frac{1}{4} \leq 0
\end{aligned}
$$

for small $L > 0$. We set the $L$ accordingly. Returning to the semi-metric, it is lower bounded as

$$
\begin{aligned}
d\big(\theta(P^{\sigma_0}), \theta(P^{\sigma_1})\big) &= \mathcal{E}_{P^{\sigma_0}}(g^{\star}_{\mu,\sigma_1}, \mu) \\
&\geq \frac{K_{\gamma,k_0}}{2} \epsilon^{\gamma_{k_0}+d} \delta(\sigma_0, \sigma_1) \\
&\geq \frac{K_{\gamma,k_0}}{2} \epsilon^{\gamma_{k_0}+d} \frac{m_0}{4} \\
&\geq \frac{K_{\gamma,k_0}}{8} \epsilon^{\gamma_{k_0}+d} K_\alpha 2^{-\alpha} K_{\gamma,k_0}^\alpha \epsilon^{\alpha\gamma_{k_0}-d} \\
&\qquad \text{(because } m_0 = \lfloor K_\alpha 2^{-\alpha} K_{\gamma,k_0}^\alpha \epsilon^{\alpha\gamma_{k_0}-d} \rfloor) \\
&= c_1 \epsilon^{(1+\alpha)\gamma_{k_0}} \\
&\geq c_2 \big\{ \mu_{k_0} n^{-\frac{\gamma_{k_0}}{2\gamma_{k_0}+d}} \big\}^{1+\alpha} = \alpha_r
\end{aligned}
$$

for some constants $c_1, c_2 > 0$. We plug in the lower and upper bound in Fano's lemma C.5 to obtain the lower bound:

$$
\frac{\alpha_r}{2}\left(1 - \frac{\beta_r+\log 2}{\log r}\right) \geq \frac{c_2\big\{\mu_{k_0} n^{-\frac{\gamma_{k_0}}{2\gamma_{k_0}+d}}\big\}^{1+\alpha}}{2} \times \frac{1}{4} \geq c_3\big\{\mu_{k_0} n^{-\frac{\gamma_{k_0}}{2\gamma_{k_0}+d}}\big\}^{1+\alpha}
$$

for some $c_3 > 0$ that is independent of both $n$ and $\mu$.

$\square$

## D  JUDGE PROMPT

Prompt format for the Llama 3.1 70b evaluator instruct LLM

```
1 <dmf>user
2 I want you to act as a judge for how well a model did answering a user
    -defined task. You will be provided with a user-defined task that
    was given to the model, its golden answer(s), and the model's
    answer. The context of the task may not be given here. Your task
    is to judge how correct is the model's answer. Your task is to
    judge how correct the model's answer is based on the golden answer
    (s), without seeing the context of the task, and then give a
    correctness score. The correctness score should be one of the
    below numbers: 0.0 (totally wrong), 0.1, 0.2, 0.3, 0.4, 0.5, 0.6,
    0.7, 0.8, 0.9, or 1.0 (totally right). You should also add a brief
     justification regarding how the model's answer conforms to or
    contradicts the golden answer(s).
```

```
3
4  Your response must follow the format
5  {{
6      "correctness_score": your_score,
7      "justification": your_justification
8  }}
9
10 Note that each one of the golden answers is considered correct. Thus
       if the Model's Answer matches anyone of the golden answers, it
       should be considered correct.
11 <dmf>assistant
12 Certainly! I will act as a judge and adhere to the format you
       requested. Under oath, I will judge without consulting my internal
        knowledge of facts under any circumstances.
13 <dmf>user
14 User-defined task --- Sandy bought 1 million Safe Moon tokens. She has
        4 siblings. She wants to keep half of them to herself and divide
       the remaining tokens among her siblings. After splitting it up,
       how many more tokens will she have than any of her siblings?
15
16 Golden Answer(s) --- \<answer 1\> 375000
17
18 Model's Answer --- Sandy will have more tokens than any sibling by 3/8
        million.
19 <dmf>assistant
20 {{
21     "correctness_score": 1.0,
22     "justification": "The golden answer states that Sandy will have
       375,000 more tokens than any of her siblings, which is a precise
       numerical value. The model's answer translates this scenario into
       a fraction of the total, saying Sandy will have more tokens than
       any sibling by 3/8 million. 1 million tokens * 3/8 =375,000 tokens
       . So the model provided an answer in fractional form that, when
       converted to a numerical value, exactly matches the golden answer'
       s quantity."
23 }}
24 <dmf>user
25 User-defined task --- Extract the names and emails mentioned in the
       following text. Richard Matthew Stallman, also known by his
       initials, rms,[1] is an American free software movement activist
       and programmer. He can be reached at rms@gnu.org based on the
       information on his webpage.
26
27 Golden Answer(s) --- {{"names":  ["Richard Matthew Stallman"], "emails
       ": ["rms@gnu.org"]}}
28
29 Model's Answer --- names: Richard Matthew Stallman
30 <dmf>assistant
31 {{
32     "correctness_score": 0.5,
33     "justification": "The model correctly identified the names but
       failed to extract the emails."
34 }}
35 <dmf>user
36 User defined task --- Question: In 2015 Edgar Lungu became prime
       minister of?
37
38 Golden Answer:
39 <answer 1> Zambia; <answer 2> Zamibia; <answer 3> People of Zambia; <
       answer4> Zambian cuisine; <answer 5> Zambians; <answer 6> Culture
       of Zambia; <answer 7> Etymology of Zambia; <answer 8> Zambia; <
       answer 9> Health care in Zambia; <answer 10> ISO 3166-1:ZM; <
       answer11> Republic Of Zambia; <answer 12> Cuisine of Zambia; <
       answer 13> Sport in Zambia; <answer 14> Republic of Zambia; <
       answer 15> Zambian people; <answer 16> Name of Zambia
```

```
40
41  Model's Answer: Prime Minister
42  <dmf>assistant
43  {{
44      "correctness_score": 0.0,
45      "justification": "The golden answers provide a detailed list of
        entities all relating to Zambia, indicating that Edgar Lungu
        became the leader (specifically, they mentioned \"prime minister
        \") of Zambia in 2015. The model's answer, \"Prime Minister,\"
        merely repeats part of the question without answering it."
46  }}
47  <dmf>user
48  User defined task --- Give a one-line concise summary of the following
        news
49  PARIS (Reuters) - French President Emmanuel Macron will set out plans
        for reforming the European Union on Tuesday, including proposals
        for a separate eurozone budget, despite a German election result
        that is likely to complicate his far-reaching ambitions. German
        Chancellor Angela Merkel s conservatives saw their support slide
        in Sunday s election, though they remain the biggest parliamentary
        bloc. She is expected to seek a coalition with the liberal Free
        Democrats (FDP) - who have criticized Macron s ideas for Europe -
        and the Greens. Elysee officials said Macron, who has promised
        sweeping reforms to Europe s monetary union in coordination with
        Merkel, hoped the issues to be raised in his speech would be taken
        into account in Germany s coalition negotiations. One Elysee
        official said a eurozone budget, one of Macron s most contentious
        ideas, would be necessary in due course and that the president
        would therefore raise the issue in his speech, to be delivered at
        the Sorbonne University in Paris. Since his election in May,
        Macron has made the overhaul of the EU and its institutions one of
        his major themes. As well as his eurozone budget idea, he wants
        to see the appointment of a eurozone finance minister and the
        creation of a rescue fund that would preemptively help countries
        facing economic trouble. Ahead of Sunday s election, Merkel had
        indicated her willingness to work with Macron on a reform agenda,
        even if her own ideas may not reach as far as his. But the
        election results have left Merkel facing a difficult coalition-
        building task which is in turn likely to limit her flexibility on
        Europe. A coalition of Merkel s CDU/CSU bloc, the FDP and the
        Greens is unprecedented at the national level - and any attempt by
        the chancellor and Macron to press for greater EU integration
        will face opposition from the new German lower house Bundestag.
        The FDP has called for a phasing out of Europe s ESM bailout fund
        and changes to EU treaties that would allow countries to leave the
        euro zone. And the far-right, eurosceptic Alternative for Germany
        is now the third biggest party in the Bundestag, further curbing
        Merkel s room for maneuver. But Elysee officials noted that the
        FDP had reaffirmed its attachment to the EU and to strong Franco-
        German relations, a point Macron was likely to emphasize in his
        speech while at the same time not seeking to impose anything on
        his partners. Macron, the sources said, would propose that the
        whole EU move forward together, and that those who did not want to
        should not stand in the way of those that did. Coming just two
        days after the German election, Macron s speech is likely to be
        interpreted in Germany as an attempt to shape the debate before
        the coalition talks begin in earnest. German coalition agreements
        are strict, with the contours set out in them limiting the
        government s room for maneuver.
50  In that respect, Merkel s ability to work with Macron on EU reform
        will be pre-determined by whatever coalition deal is struck. FDP
        leader Christian Lindner said on Monday he would not agree to any
        coalition that did not promise a change in the German government s
        direction. While that appeared to set the stage for tough talks,
        he also offered hope for Macron, saying when asked about the
```

```
     French president s eurozone budget ideas that the FDP had a strong
      interest in the strength of France.
51
52 Golden Answer(s) --- After German election, Macron to set out his
      vision for Europe
53
54 Model's Answer --- French President Emmanuel Macron to introduce plans
       for reforming the European Union amid the uncertain aftermath of
      German elections
55 <dmf>assistant
56 {{
57     "correctness_score": 0.6,
58     "justification": "While the golden answer is more concise, the
      model's answer is largely similar to the golden answer in its
      meaning."
59 }}
60 <dmf>user
61 User defined task --- {MODELTASK}
62 Golden Answer(s) --- {GOLDENRESPONSE}
63 Model's Answer --- {MODELRESPONSE}
```

