# OpenReview forum: "CARROT: A Cost Aware Rate Optimal Router"
_ICLR.cc/2026/Conference — ICLR 2026 Conference Withdrawn Submission_

### Official Review · Reviewer_4zBY · 2025-10-29

**Soundness:** 2
**Presentation:** 3
**Contribution:** 2
**Rating:** 2
**Confidence:** 4

**Summary:**

- Formulates LLM routing as a minimax learning problem, deriving lower and upper bounds on excess risk and proving that a plug-in router predicting both cost and accuracy is rate-optimal.
- Introduces CARROT, an implementation of this theoretical plug-in router that selects the cheapest model expected to meet performance goals.
- Releases SPROUT, a new dataset (~45 k prompts × 14 LLMs) including per-query cost, accuracy, and chat-template formatting for studying cost-aware routing.
- Empirically shows CARROT achieves similar accuracy as GPT-4-level models at much lower cost, outperforming prior routers such as RouteLLM and RouterBench.

**Strengths:**

the paper provides a clear mathematical framework for the routing problem with a principled treatment of routing - a increasingly practical relevant problem

**Weaknesses:**

- Limited novelty. The main idea of routing queries between models based on cost or accuracy is well established. The theoretical analysis adds formality but does not lead to a substantively new method or insight.
- Model–prompt coupling not addressed. In practice, system prompts are heavily engineered for specific models. Swapping models without re-tuning instructions is rarely valid, and the mild adjustments through per-model chat templates seem insufficient to ensure fair comparison or practical relevance.
- Non-deterministic model behavior ignored. LLM outputs can vary substantially across runs, yet the paper assumes deterministic outputs and evaluates each (prompt, model) pair only once. This undermines the realism of both the theoretical and empirical results.
- Unclear link between “accuracy” and task performance. It is not evident how the reported benchmark accuracy maps to real task success. Many queries have no single correct answer, and performance may depend on stylistic or preference factors. If the authors addressed this, it was not clearly explained.
- Zero-shot prompting assumption. The authors frame zero-shot prompting as a strength, but in practice few-shot or instruction-tuned prompting is essential for good performance. This limits the applicability of the results to real deployment settings.
- Incomplete literature coverage and outdated baselines. The paper is not the first to consider cost-aware routing. Prior work and other dynamic routing frameworks address similar questions with more advanced or realistic approaches.

**Questions:**

what do the authors mean by "Unlike previous benchmarks, it does not rely on few-shot prompting"? Few shot prompting is critical in many cases to get good task performance. In fact, different models prefer different type of few shot prompting (along the lines of system prompts are adapted for the model)

---

### Official Review · Reviewer_Avya · 2025-10-30

**Soundness:** 3
**Presentation:** 3
**Contribution:** 3
**Rating:** 4
**Confidence:** 3

**Summary:**

This paper studies an interesting problem of LLM routing. It first conducts a minimax analysis of the routing problem, providing a lower bound and some theoretical insights. Furthermore, this work introduces CARROT method that routes the LLM based on the estimates of the models' cost and performance with a weighted vector. And this work also introduces a new LLM routing evaluation dataset named SPROUT. Experimental results indicate the strong performance of CARROT over existing baselines.

**Strengths:**

1. I feel LLM routing is a very interesting and practical problem.
2. This work contains both theoretical analysis and empirical results, which look complete to me.
3. The empirical result show the strong performance of the proposed CARROT algorithm over baselines on several benchmarks.
4. I found the presentation of this work clear, and most details are included in the Appendix.

**Weaknesses:**

1. I feel the novelty of the theoretical deduction is overall limited. By reading the proof, I feel the analysis is adapted from the existing statistical learning theory on binary classification and risk control. And there has been quite comprehensive existing theoretical literature on lower bound and upper bound deduction for the problem setting used in this work. Could authors elaborate on any novelty on your theory?
2. I feel this work can include more baselines for comparison as there are some recent works on LLM routing, such as Universal Routing. Otherwise, the empirical results may not be very comprehensive and up-to-date.
3. It is also nice to see the out-of-distribution performance of the proposed CARROT algorithm, as it is more interesting and has been studied by recent works. Intuitively, both KNN and the trained Roberta model may suffer a lot from the out-of-distribution testing sets.

**Questions:**

Please refer to the above Weaknesses section.

---

### Official Review · Reviewer_Yw5u · 2025-11-03

**Soundness:** 3
**Presentation:** 3
**Contribution:** 2
**Rating:** 4
**Confidence:** 4

**Summary:**

This paper proposes CARROT, an LLM router designed to select the optimal model based on a cost-performance trade-off for each individual query. Addressing the gap that prior work often assumes static costs, CARROT implements a "plug-in" approach that dynamically estimates both the performance and the inference cost for each query using two types of estimators (KNN and Roberta). The authors provide a minimax analysis to theoretically prove this plug-in method is statistically rate-optimal. To support this, the paper also introduces the SPROUT dataset, a new benchmark for cost-aware routing that uses an LLM-as-judge for evaluation. The system's empirical performance is then validated against several baselines on SPROUT and other existing benchmarks.

**Strengths:**

1. The paper effectively demonstrates the empirical value of its approach. The experiments show that a cost-aware router like CARROT can achieve a more efficient cost-performance trade-off than baselines, particularly binary routers and non-predictive routers.
2. The paper provides a theoretical justification for its method in Section 3 . While the "plug-in" concept of estimating and then optimizing is intuitive, the authors formally prove that this simple approach is, in fact, minimax rate-optimal given an optimal estimator. This provides a statistical foundation for the design.

**Weaknesses:**

1. The structure of the introductory sections feels somewhat unconventional. The second paragraph is very long. It would be clearer to split this section into "Introduction" and "Related Work/Background". Additionally, the current Section 2 and Section 3 are intrinsically linked, defining and then analyzing the same problem. These could be combined into a single, comprehensive "Problem Formulation and Theoretical Analysis" section to improve readability.
2. A concern arises from the construction of the SPROUT dataset. The authors use an LLM (LLaMa-3.1-70B-Instruct) as a judge to evaluate response correctness. This introduces potential for unverified bias or error. How can the authors ensure the judge's evaluations are correct and reliable? The paper would be stronger if it included an analysis of the judge's reliability. This also raises the question of why this subjective method was used instead of designing tasks that enforce a fixed output format, which would permit objective, automated evaluation.
3. This is my primary concern. The paper's main contributions are (1) the theory that estimating metrics is sufficient, and (2) the practice of training two estimators (KNN and Roberta) to do so. The experimental evaluation (Figures 2 & 3) conflates these two contributions by only showing the final end-to-end performance. The resulting Pareto frontier shape is an expected and common result in multi-objective convex optimization (e.g., when solving problems like $\min_x \mu ||Ax - b||^2 + (1-\mu) ||x||^2$ by sweeping $\mu$). The evaluation is missing a crucial part: a direct assessment of the estimators' performance. The paper must include metrics showing how accurate the KNN and Roberta models are at predicting the ground-truth cost and performance scores (e.g., by reporting the prediction accuracy).
4. I wonder if the trained estimators can generalize to out-of-distribution (OOD) queries. The evaluation appears to be on standard in-domain splits. This raises a question: how can simple models like KNN or roberta-base truly understand the semantic difficulty of a query? For example, a query like "Prove Fermat's Last Theorem" is syntactically simple but semantically profound (and actually from Fermat's proposal to Wiles' proof took over 300 years). It is unclear how these estimators would differentiate it from a trivial query and correctly estimate the massive cost/performance required, suggesting a potential failure mode for complex, unseen problems.

**Questions:**

Please see weaknesses part above.

---

### Official Review · Reviewer_K3pK · 2025-11-03

**Soundness:** 3
**Presentation:** 3
**Contribution:** 3
**Rating:** 6
**Confidence:** 4

**Summary:**

This paper introduces CARROT, a cost-aware rate optimal router that routes user queries across LLMs based on estimated model costs and performance. Authors also introduce a new benchmark, SPROUT, to facilitate routing on a wide spectrum of queries. Extensive experiments on SPROUT and other benchmarks demonstrate the efficiency and effectiveness of the proposed CARROT approach.

**Strengths:**

1. This paper formulates the LLM routing problem as a minmax rate optimal problem. The technical development seems solid.
2. How to balance the achieved response quality and incurred inference cost is a critical problem in modern LLM serving system.
3. The evaluation result is impressive. For example, at 30% of the cost, CARROT matches or exceeds the performance of GPT-4o on each benchmark.

**Weaknesses:**

1. Some advanced routing baselines are neither compared nor discussed. For example,
    - Ding, Dujian, et al. "BEST-Route: Adaptive LLM Routing with Test-Time Optimal Compute." Forty-second International Conference on Machine Learning.
2. This paper leverages a sequence of assumptions. It would add great values to this work if authors could discuss to which extent these assumptions may hold in practice.

**Questions:**

Can CARROT be extended to multi-turn conversation -- the mainstream LLM serving scenarios? Typically, due the the carried-on contexts, the LLM costs (e.g., KV cache transfer or re-compute) may vary as the conversation progresses and leads to new challenges.

---

### Official Review · Reviewer_n1HQ · 2025-11-07

**Soundness:** 2
**Presentation:** 2
**Contribution:** 1
**Rating:** 2
**Confidence:** 3

**Summary:**

The paper introduces CARROT (Cost AwaRe Rate Optimal rouTer), a theoretically grounded and empirically validated framework for cost-aware routing among large language models (LLMs). While prior routing methods either ignore query-dependent cost variations or rely on costly multi-model inference cascades, CARROT achieves minimax-optimal efficiency by jointly predicting both accuracy and inference cost for each candidate model. The authors first present a minimax analysis of the routing problem, deriving a lower bound on excess risk and proving that a plug-in router—estimating conditional expectations of model metrics given a query—can asymptotically attain this bound. Building on this insight, CARROT implements a two-stage plug-in approach: it learns predictors for model-wise accuracy and cost from training data, then selects the model minimizing a convex combination of the two estimated risks. To support realistic training and evaluation, the authors introduce SPROUT (Smart Price-aware ROUTing), a large benchmark covering 14 state-of-the-art LLMs and over 45k prompts from reasoning, retrieval-augmented generation, and general instruction datasets.

**Strengths:**

1. The performance of CARROT is competitive.

**Weaknesses:**

1. For better or worse, the paper is heavily packed with formal theorems and minimax analyses. While this mathematical framing is thorough, it sometimes overshadows the practical contribution. It would be helpful if the authors clearly indicated where the proofs for each theorem are located.
2. What's the difference between CARROT and prior routing work, i.e. the main novelty here? I believe similar multi-model risk estimators have been used in earlier “model selection” and “LLM routing” papers, e.g. RouteLLM.
3. Although SPROUT is introduced as a new benchmark, it largely repackages existing benchmarks—specifically GPQA, MuSR, MMLU-Pro, MATH, OpenHermes, and RAGBench—rather than creating new data or evaluation protocols. Its construction mainly involves aggregating prompts and responses from prior datasets under a unified cost annotation framework, which limits its originality.

**Questions:**

Please see Weaknesses.

---

### Note · Authors · 2025-11-14

I have read and agree with the venue's withdrawal policy on behalf of myself and my co-authors.